# Advancing knock-in approaches for robust genome editing in zebrafish

Anjelica Rodriguez-Parks[1], Ella Grace Beezley[1], Steffani Manna[1], Isabella Silaban[1], Sarah I. Almutawa[1], Siyang Cao[1], Hossam Ahmed[1], Megan Guyer[1], Sean Baker[2], Mark P. Richards[2] and Junsu Kang[1,3,*]

## ABSTRACT

Precise genome editing remains a major challenge in functional genomics, particularly for generating knock-in (KI) alleles in model organisms. Here, we introduce the mini-golden system, a versatile Golden Gate-based subcloning platform that enables rapid assembly of donor constructs containing homology arms and a gene of interest. This system offers a library of middle entry vectors including diverse genes, enhancing the preparation of donor minicircles for KI applications. Using the mini-golden system, we efficiently generated a *foxd3^CreER* KI zebrafish line, allowing conditional recombination in neural crest cells. To further improve genome editing precision, we developed a synthetic exon-based donor template strategy combined with fluorescence screening. Using this approach, we successfully engineered a targeted isoleucine-to-valine substitution (Ile-to-Val) in *hbaa1.2*, one of the two adult hemoglobin alpha genes in zebrafish. Importantly, despite the high sequence similarity between *hbaa1.2* and its paralog *hbaa1.1*, our strategy specifically edited *hbaa1.2*, demonstrating the effectiveness of the synthetic exon approach. This method minimized undesired recombination and significantly improved the identification of lines carrying the edited genome. Together, we provide a robust toolkit for efficient and precise genome engineering in zebrafish, with broad applicability to other model systems.

KEY WORDS: Knock-in, Genome editing, Zebrafish, Amino acid substitution, Foxd3, Hbaa

## INTRODUCTION

The advent of genome editing techniques, such as CRISPR/Cas9, has revolutionized genome editing approaches, allowing for targeted knock-out (KO) and knock-in (KI) applications (Hwang et al., 2013; Auer et al., 2014; Sander and Joung, 2014). KOs, such as mutating or deleting DNA fragments, are relatively easy, compared to KIs, which require more intricate and precise steps

[1]Department of Cell and Regenerative Biology, School of Medicine and Public Health, University of Wisconsin-Madison, Madison, WI 53705, USA. [2]Department of Animal and Dairy Sciences, University of Wisconsin-Madison, Madison, WI 53706, USA. [3]UW Carbone Cancer Center, School of Medicine and Public Health, University of Wisconsin-Madison, Madison, WI 53705, USA.

*Author for correspondence ( junsu.kang@wisc.edu)

A.R.-P., 0009-0001-0748-4146; S.M., 0009-0002-1209-2849; I.S., 0009-0002-7349-7903; S.I.A., 0000-0003-4104-4280; S.C., 0009-0008-7880-841X; H.A., 0009-0008-8205-4139; M.G., 0009-0008-6287-7664; S.B., 0000-0001-6432-2277; M.P.R., 0000-0003-1152-9420; J.K., 0000-0001-5286-5426

(Hoshijima et al., 2019; Prykhozhij and Berman, 2024). KI strategies typically involve providing a donor template containing the gene of interest flanked by homology sequences at 5′ and 3′ regions of the target site. When this donor template is delivered into cells along with Cas9 and sgRNA, either homology-directed repair (HDR) or nonhomologous end joining (NHEJ) mechanism facilitates the integration of the gene of interest at the target site. Although KI strategies have been substantially optimized, editing efficiency remains variable and often low, typically ranging from 2-40% (Auer et al., 2014; Ata et al., 2016; Burg et al., 2018; Gutierrez-Triana et al., 2018; Prykhozhij et al., 2018; Wierson et al., 2020; Almeida et al., 2021; Levic et al., 2021; Mi and Andersson, 2023; Zhang et al., 2023; Prykhozhij and Berman, 2024; Oikemus et al., 2025). Moreover, KI for base editing generally relies on PCR-based screening to identify individuals carrying the edited genome (Rosello et al., 2022; Zheng et al., 2023; Qin et al., 2024; Liu et al., 2025; Zhong et al., 2025). However, this PCR-based screening is labor-intensive, time-consuming, and may fail to detect rare but correctly targeted events. Improving the efficiency and reliability of screening strategies is therefore essential for maximizing the practical utility of KI-based genome editing approaches.

We and other groups demonstrated that minicircles can serve as a donor template enhancing the KI efficiency (Suzuki et al., 2016; Keating et al., 2024). However, creating a donor template containing both homology arms (HAs) is laborious and time-consuming. Generating a donor construct for KI generation requires multiple steps to add 5′ and 3′ HAs and supplementary sequences, such as a selection marker, depending on the complexity of the plasmids. Here, we developed the mini-golden system, a convenient, efficient subcloning platform and plasmid library designed to boost preparation of diverse donor templates. Utilizing this mini-golden system, we successfully generated the *foxd3^CreER* KI line, which allows conditional recombination in neural crest cells (NCCs). The mini-golden system allows for one-step customization of donor templates, providing a robust and versatile tool for genome editing.

The genetic similarity between zebrafish and human positions zebrafish as an invaluable model for studying human diseases (Howe et al., 2013). Recent advancement of base and prime editing techniques enables the introduction of human disease-causing mutations into the zebrafish genome (Rosello et al., 2021; Petri et al., 2022; Richardson et al., 2023), significantly promoting the application of zebrafish for human disease modeling. Furthermore, iterative improvements in base editing, including optimization of the editing window, expansion of the PAM compatibilities, and introducing new DNA binding domains, have increased flexibility and broadened the range of targetable loci (Rosello et al., 2022; Zheng et al., 2023; Qin et al., 2024; Liu et al., 2025; Zhong et al., 2025). Despite these advances, the typical editing window for base editing spans several base pairs, creating the possibility of unintended edits when identical nucleotides occur within that

window. Moreover, base editing approaches are inapplicable when identical sgRNA target sites are present at other genomic loci, due to the increased risk of off-target editing. Also, the current base and prime editing approaches rely on PCR-based genotyping to identify lines carrying the edited genome, which introduces multiple procedural steps and increases the risk of missing true positives. To improve screening efficiency, we incorporate a fluorescence selection marker, greatly advancing the identification of positive larvae through simple microscopy. Additionally, we employ a synthetic exon approach to avoid undesired recombination, further increasing efficiency. By integrating a synthetic exon strategy with a robust fluorescence-based screening method, we establish a highly effective approach for introducing amino acid (a.a.) changes in zebrafish models. Our strategy offers valuable resources and methodology to the genome editing toolbox for zebrafish and other animal model communities.

## RESULTS

### mini-golden system for convenient and prompt subcloning

The mini-golden system utilizes Golden Gate assembly (Engler et al., 2008) to simultaneously and directionally combine 5′ HA, a gene of interest, and 3′ HA (Fig. 1A, Fig. S1A). The mini-golden

system consists of four components: 5′ and 3′ HAs, a middle entry vector (pMC-ME), and a destination vector (pMC-Dest). Our mini-golden system uses the BsaI restriction enzyme to create 4-base overhangs, thus all four components must not carry additional BsaI sites. 5′ and 3′ HAs are prepared by PCR from genomic DNA followed by gel extraction purification. Any internal BsaI sites in HAs must be mutated via overlap extension PCR or other available methods. For the destination vector, we modified the previously available minicircle generating vector[17] by mutating existing BsaI sites and adding two outward facing BsaI sites in the multi-cloning site. We also constructed multiple pMC-MEs encoding several fluorescence proteins (EGFP, mCherry, BFP, mNeongreen, mScarlet), membrane localized fluorescence proteins, CreER, or NTRv2 (Sharrock et al., 2022) (Fig. 1B, Table S1). The detailed protocol for mini-golden system is described in the Materials and Methods. The mini-golden system greatly facilitates quick and convenient subcloning to prepare a donor minicircle for KI strategy.

### Generation of *foxd3^CreER^* KI strain via the mini-golden system

*Cre/loxp* system is a widely used genetic tool for spatiotemporal control of gene expression and lineage tracing (Sauer and Henderson, 1988; Liu et al., 2022). CreER, a fusion protein of Cre recombinase

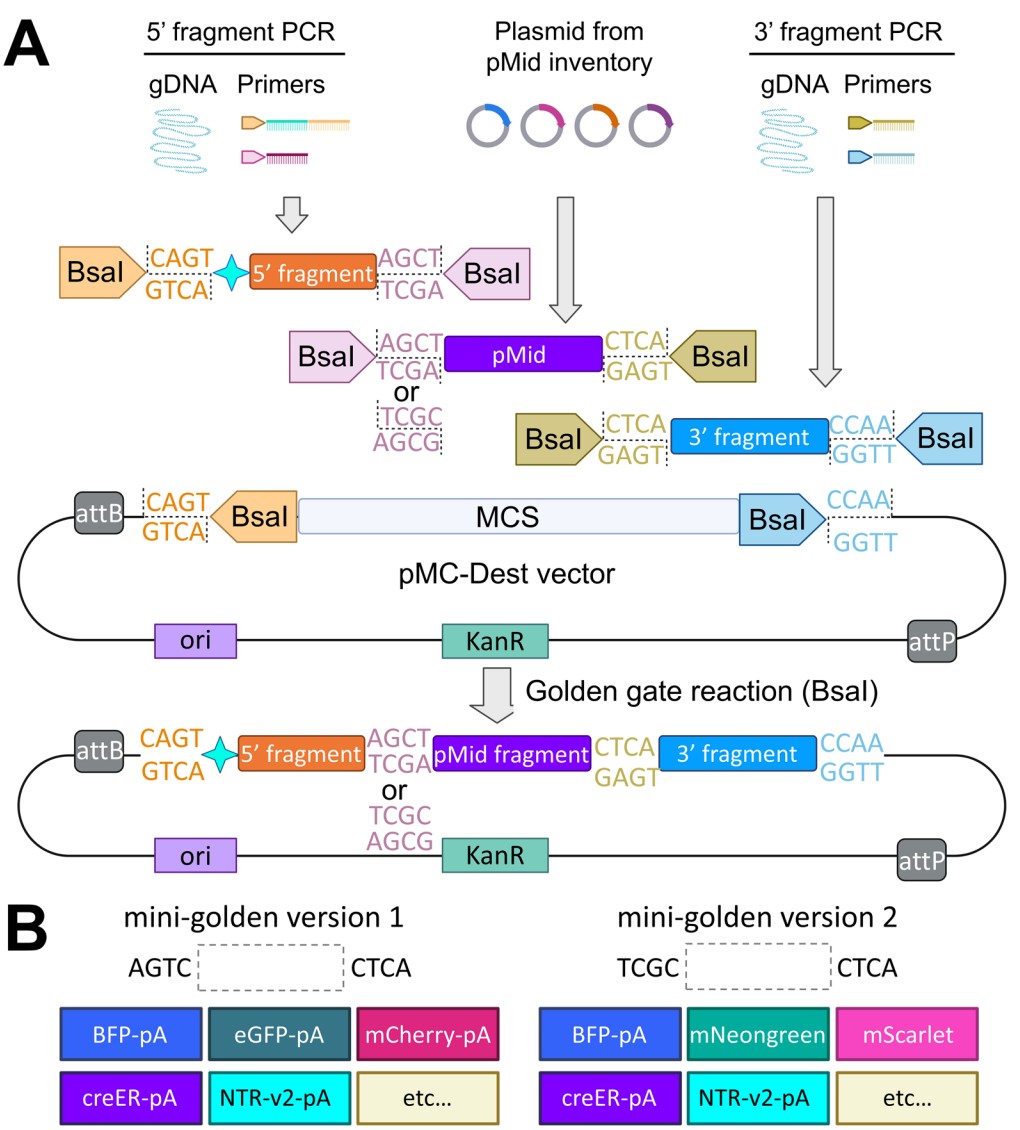

Fig. 1. mini-golden system for robust creation of donor templates. (A) The schematic of the mini-golden system. The 5′ and 3′ HA fragments are prepared by PCR amplification followed by gel purification. Middle entry vector is chosen from the pMC-ME library. Linker sequences for mini-golden version 1 (Addgene deposit 82577) and version 2 (Addgene deposit 86145) are ACGT-CTCA and TCGC-CTCA, respectively. (B) pMC-ME middle entry vector library contains diverse constructs. The detailed protocol for mini-golden system is described in the Materials and Methods.

and the estrogen receptor (ER) ligand binding domain, enables temporal regulation through inducible activation. To generate a NCC *CreER* driver, we targeted *foxd3*, a pivotal transcription factor essential for NCC specification and differentiation (Lister et al., 2006; Stewart et al., 2006; Curran et al., 2009; Hochgreb-Hagele and Bronner, 2013; Candido-Ferreira et al., 2023), using our mini-golden system.

We designed our strategy to integrate *CreER* into the 5′ untranslated region (UTR) of *foxd3* and to delete a portion of the *foxd3* coding sequence to generate deletion mutants (Fig. 2A, Fig. S1B). The sgRNA site is located approximately 830 bp upstream of the *foxd3* start codon. The 5′ HA fragment spans 826 bp starting from the sgRNA site, and the 3′ HA covered 547 bp beginning 403 bp downstream of the 5′ HA. From our mini-golden middle-entry library, we selected a plasmid containing *CreER* paired with an *FRT*-flanked *α-cry:mCherry* selection marker, which facilitates the identification of larvae carrying the modified genome. If necessary, *α-cry:mCherry* can be removed by introducing *flp* recombinase mRNA to prevent interference with *CreER* expression (Sugimoto et al., 2017). These components were assembled into a destination vector to create the donor plasmid. The final donor DNA was prepared using minicircle technology (Kay et al., 2010), which produces a minicircle DNA containing necessary DNAs without the bacterial backbone sequences (Fig. S1A). Our previous work demonstrated that

minicircle significantly enhances KI efficiency compared to a plasmid (Keating et al., 2024).

We co-injected the donor minicircle, sgRNA, and Cas9 into one-cell stage embryos and assessed genome editing efficiency. The rate of $\alpha$-cry:mCherry$^+$ $F_0$ larvae was 60.37% (99/164). Among $\alpha$-cry:mCherry$^+$ $F_0$, 2 out of 16 individual embryos exhibited the correct 5′ and 3′ junction sizes (Fig. S1C). $\alpha$-cry:mCherry$^+$ $F_0$ larvae were grown to adult, and we screened lens mCherry expressing fish again at the adult stage. From ∼30 $\alpha$-cry:mCherry$^+$ $F_0$ adult, we established seven lines transmitting $\alpha$-cry:mCherry$^+$ to the next generation (Fig. 2B-D, Fig. S1D). PCR analysis of $F_1$ animals revealed that four lines failed to amplify both 5′ and 3′ flanking regions. Line 8 showed the expected 3′ fragment but a longer 5′ fragment, while line 9 produced the expected 5′ fragment but a longer 3′ fragment (Fig. 2B,D, Fig. S1D). Line 23 generated correctly sized bands at both junctions (Fig. 2C,D). Sanger sequencing analyses confirmed that *CreER* was integrated into the *foxd3* locus via NHEJ in line 9, hereafter referred to as *foxd3$^{CreER-NH}$*, (Fig. 2B,D). By contrast, line 23 exhibited HDR-mediated integration, hereafter referred to as *foxd3$^{CreER-HR}$*, (Fig. 2C,D).

*foxd3* is an important transcription factor for pre-migratory NCC specification and fate determination, and *foxd3* mutants display deformed jaws (Lister et al., 2006; Stewart et al., 2006; Hochgreb-Hagele and Bronner, 2013). To assess whether *CreER* integration

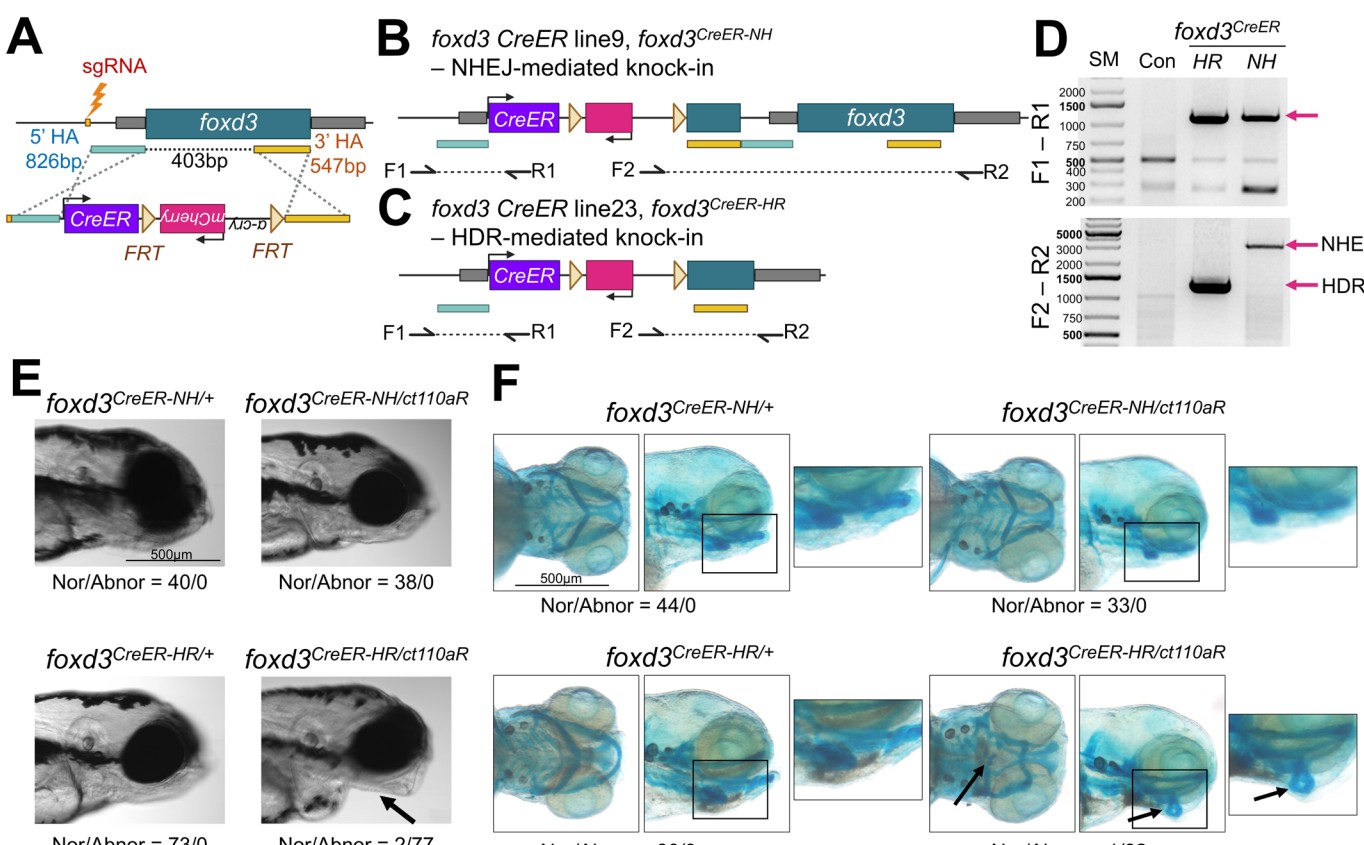

**Fig. 2. Generation of *foxd3CreER* KI line.** (A) Schematic of genome editing strategy to create foxd3CreER KI line. (B,C) Schematic of *foxd3$^{CreER-NH}$* (B) and *foxd3$^{CreER-HR}$* (C). *foxd3$^{CreER-NH}$* was created via non-homologous end joining (NHEJ)-mediated repair, resulting in an intact *foxd3* gene sequence. In contrast, *foxd3$^{CreER-HR}$* was generated via HDR, leading to deletion of the 5′ portion of *foxd3*. (D) PCR analysis to genotype upstream (F1-R1) and downstream (F2-R2) regions of the integrated sites at the *foxd3* locus. (E) Lateral view of control heterozygotes and compound heterozygotes of the *foxd3$^{CreER-NH}$* or *foxd3$^{CreER-HR}$* and the *foxd3* enhancer trap line (ct110aR) at 4 dpf. While *foxd3$^{CreER-HR/+}$*, *foxd3$^{CreER-NH/+}$* and *foxd3$^{CreER-NH/ct110aR}$* develop jaws normally, *foxd3$^{CreER-HR/ct110aR}$* exhibits craniofacial defects (arrow). Nor, normal; Abnor, abnormal. (F) Ventral and lateral views of pharyngeal cartilages stained with Alcian Blue at 5 dpf. Pharyngeal cartilages and arches are intact in *foxd3$^{CreER-HR/+}$*, *foxd3$^{CreER-NH/+}$* and *foxd3$^{CreER-NH/ct110aR}$*, whereas *foxd3$^{CreER-HR/ct110aR}$* exhibits disrupted cartilage phenotype (arrow). Scale bars: 500 µm. Panels A, B and C were created in BioRender by Kang, J., 2026. https://BioRender.com/ycswmq4, https://BioRender.com/ly0bf4a and https://BioRender.com/cqdln93, respectively. This figure was sublicensed under CC-BY 4.0 terms.

Biology Open

affects *foxd3* function, we crossed *foxd3*<sup>CreER-NH</sup> or *foxd3*<sup>CreER-HR</sup> with G*t(foxd3:mcherry)*<sup>ct110aR</sup>, a *foxd3* gene trap reporter and loss-of-function allele (Hochgreb-Hagele and Bronner, 2013). While *foxd3*<sup>CreER-NH/ct110aR</sup> exhibited no craniofacial abnormalities, we observed that *foxd3*<sup>CreER-HR/ct110aR</sup> displayed jaw malformations (Fig. 2E). All heterozygote controls have normal shaped jaws. Alcian Blue staining revealed the defects in the posterior pharyngeal elements of *foxd3*<sup>CreER-HR/ct110aR</sup>, but not in *foxd3*<sup>CreER-NH/ct110aR</sup> and heterozygote controls (Fig. 2F). Moreover, reverse transcription polymerase chain reaction (RT-PCR) analysis confirmed that functional *foxd3* transcripts were undetectable in cDNA extracted from *foxd3*<sup>CreER-HR/ct110aR</sup>, whereas *foxd3*<sup>CreER-NH/ct110aR</sup> has it (Fig. S2). Collectively, our mini-golden-mediated strategy successfully integrated *CreER* into the *foxd3* locus. *foxd3*<sup>CreER-HR</sup> represents a complete *foxd3* loss-of-function allele, while *foxd3*<sup>CreER-NH</sup> retains the endogenous *foxd3* activity. Because *foxd3*<sup>CreER-HR</sup> fulfilled the objective of generating both a *foxd3 CreER* driver and a loss-of-function allele, *foxd3*<sup>CreER-HR</sup> was selected for subsequent validation of Cre-mediated recombination in the NCC lineage.

### *foxd3*<sup>CreER</sup> and tamoxifen-mediated lineage tracing strategy labels neural crest-derived cells

To determine *foxd3* expression pattern, we analyzed the G*t(foxd3:mcherry)*<sup>ct110aR</sup> gene trap line (Hochgreb-Hagele and Bronner, 2013). Consistent with previous reports (Hochgreb-Hagele and Bronner, 2013), our confocal imaging confirmed mCherry expression in cranial NCCs and peripheral glial cells (Fig. S3A,B). Additionally, as shown in previous studies, we observed mCherry expression in the paraxial mesoderm starting at 2 days post-fertilization (dpf), which declined by 5 dpf (Fig. S3A,B) (Odenthal and Nusslein-Volhard, 1998; Lee et al., 2006; Stewart et al., 2006). While the *α-cry:mCherry* selection marker drives lens mCherry expression in both *foxd3*<sup>CreER-NH</sup> and *foxd3*<sup>CreER-HR</sup>, *foxd3*<sup>CreER-HR</sup> exhibits additional mCherry expression in the paraxial mesoderm (Fig. S3C,D). This unexpected expression pattern of the selection marker indicates the presence of the potential mesoderm regulatory elements downstream of *foxd3*, which may influence *α-cry:mCherry* transcription for the *foxd3*<sup>CreER-HR</sup>. *flp* recombinase mRNA injection into the one-cell stage embryos successfully delete the *FRT*-franked *α-cry:mCherry* cassette, resulting in the lack of lens expression (Fig. 3, Fig. S3E,F).

To validate whether newly generated *foxd3*<sup>CreER-HR</sup> can drive expression of functional *CreER* in NCCs, we crossed *foxd3*<sup>CreER-HR</sup> with *ubi:switch* (Mosimann et al., 2011) and examined whether NCCs were specifically labelled following tamoxifen treatment. *flp* mRNA was injected at the one-cell stage embryos to prevent *CreER*-mediated mCherry expression. 4-HT was treated from 5 to 77 h post-fertilization (hpf) to induce recombination and mCherry expression was evaluated to determine *CreER* expressing cells. At 5 dpf, mCherry expression was detectable in cranial NCCs and peripheral glial cells, matching to that of G*t(foxd3:mcherry)*<sup>ct110aR</sup> (Fig. 3 and Fig. S3A,B). These data indicate successful *CreER*-mediated-recombination of *foxd3*<sup>CreER-HR</sup> in NCCs to label NCC-derived cells.

### Adult hemoglobin alpha chain gene structures in zebrafish

Hemoglobin (Hb), a key oxygen-transporting protein in red blood cells, is composed of two alpha- and two beta-globin chains (Chan et al., 1997; Brownlie et al., 2003). While Hbs across animal species share a similar structural architecture, variations in certain a.a. residues lead to differences in oxygen-carrying capacity. For instance, fetal Hbs exhibit a higher affinity for oxygen than adult

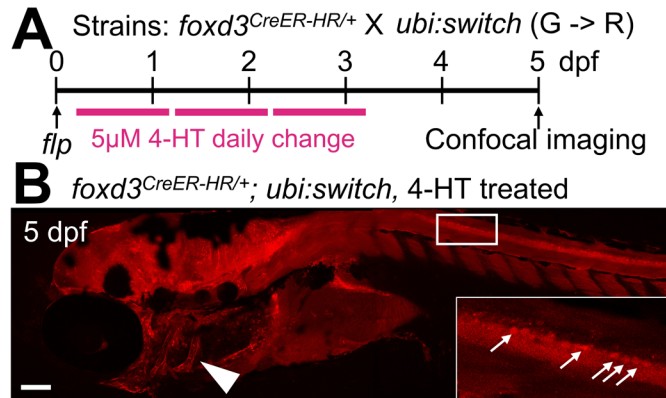

**Fig. 3. *foxd3*<sup>CreER-HR</sup> KI induces recombination in NCCs.** (A) Schematic of 4-hydroxytamoxifen (4-HT) treatment with foxd3CreER-HR/+; ubi:switch (Green to Red). *flp* mRNA was injected to remove the FRT-franked *α-cry:mCherry* cassette. (B) Representative confocal images of zebrafish larvae at 5 dpf following 4-HT treatment. The *ubi:switch* transgene undergoes recombination in NCCs upon 4-HT-induced activation of *CreER* in *foxd3*-expressing NCCs, labeling craniofacial NCC-derived cells (arrowhead) and peripheral glial cells (arrow). Insets show magnified views of peripheral glial cells in 4-HT-treated larvae (B). Scale bar: 100 μm.

Hbs (Manning et al., 2020). In most fish Hbs, an isoleucine (Ile) residue occupies position 11 of the E-helix (E11), whereas mammalian Hbs have smaller amino acids at this position, such as valine (Val) (Fig. S4A). Previous studies suggest that IleE11 may promote the dissociation of the neutral superoxide radical, potentially enhancing Hb oxidation (Aranda et al., 2009). To investigate this, we aim to substitute IleE11 to ValE11 in zebrafish.

Zebrafish genome encodes two adult alpha-globin genes (Brownlie et al., 2003), including *hbaa1* and *si:ch211-5k11.8,* which we have annotated as *hbaa1.1* and *hbaa1.2*, respectively (Fig. 4A). Both genes carry Ile at the E11 (Fig. S4A,B). To determine the major adult isoform, we collected blood from adult fish and performed mass spectrometry. Although peptides from both Hbaa1.1 and Hbaa1.2 were detected, Hbaa1.2 showed slightly higher expression levels (Fig. S4C,D). Based on this finding, we selected Hbaa1.2 for modification of the IleE11 to ValE11 residue (ItoV).

### Synthetic exon-based genome editing combined with fluorescence screening enables a single amino acid substitution in Hbaa1.2

*hbaa1.1* and *hbaa1.2* exhibit extremely high sequence identity at the a.a., coding, and genomic DNA levels (Fig. 4B). Although most sgRNA target sites are shared between the two genes, we identified an efficient sgRNA site specific to *hbaa1.2*, which is positioned 63 base pairs upstream of ATC (I64) codon. This site exhibits two nucleotide differences relative to the corresponding region of *hbaa1.1* (Fig. 4C,D). Because the sgRNA site is distal to the target IleE11 codon, the base editing approach is inapplicable for introducing the Hbaa1.2 ItoV substitution.

Another challenge for a.a. substitution is the low efficiency of selecting animals that carry the desired modification. Previous a.a. editing approaches, including single oligo-mediated and prime/base editing, relied on PCR-based screening to identify animals carrying the edited genome (Rosello et al., 2021, 2022; Petri et al., 2022; Richardson et al., 2023). Given the low efficiency of genome editing techniques and a reduced germline transmission rate, this PCR-based approach may fail to select positive lines due to the complicated multiple steps for genotyping and dilution of the edited

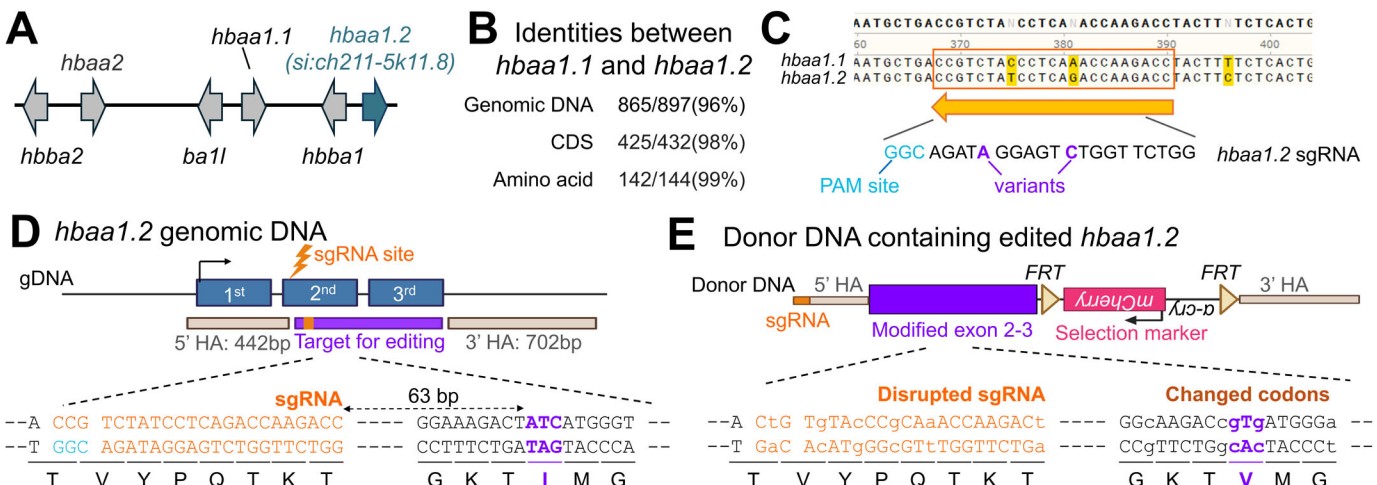

**Fig. 4. Design of a single a.a. substitution in Hbaa1.2.** (A) Schematic diagram of the α- and β-globin gene clusters in zebrafish. (B) Sequence similarity between *hbaa1.1* and *hbaa1.2* at genomic DNA, coding sequence (CDS), and a.a. level. (C) sgRNA target site in *hbaa1.2* and the homologous sequence in *hbaa1.1*. Nucleotide variants between these two genes are highlighted by red. (D) *hbaa1.2* gene structure, sgRNA sequence (orange), and the codon encoding I residue targeted for the substitution (ATC, red). Corresponding a.a. sequences are shown below each codon. (E) Structure of the donor minicircle and the synthetic exon sequence designed for *hbaa1.2* editing. Individual codons, except the one encoding the target valine (V) residue, were replaced with synonymous codons to maintain protein sequence while preventing re-cutting by Cas9 as well as undesired recombination. Panels D and E were created in BioRender by Kang, J., 2026. https://BioRender.com/av0kg50 and https://BioRender.com/rgqflwj, repectively. This figure was sublicensed under CC-BY 4.0 terms.

genome by massive amount of unedited wild-type genomic DNA. Additionally, this process is labor-intensive and time-consuming. To address these limitations, we incorporated a fluorescence selection marker to simplify screening process. This approach allows for the identification of potential positive lines using fluorescence microscopy (Fig. 5A). We used *FRT*-flanked *α-cry:mCherry*, which drives lens expression, as a selection marker (Fig. 4E). If a selection marker interferes with *hbaa1.2* gene expression, it can be eliminated by introducing the *flp* recombinase mRNA.

The target a.a. is located within the 2nd exon (Fig. 4D) and a selection marker must be placed in a non-coding region to avoid disrupting *hbaa1.2* expression. To address these challenges, we positioned the selection marker downstream of *hbaa1.2* (Fig. 5B, Fig. S5A). However, this configuration can lead to undesired recombination events between one of HA and DNA sequences surrounding the target a.a. To prevent unwanted recombination, we devised the use of a synthetic exon as a donor template. The sgRNA site and downstream coding DNA, except I64 V in the donor, are altered with synonymous substitution to minimize undesired recombination (Fig. 4E, Fig. S5A). However, the 3′ UTR region was left unmodified to retain potential regulatory elements. Taking these considerations, our donor template contains the following elements in order: a sgRNA site, a 442 bp 5′ HA, a modified partial 329 bp *hbaa1.2* CDS having I64 V change, 279 bp 3′ UTR and pA, an FRT-flanked lens-specific selection marker, and a 702 bp 3′ HA (Figs 4E, 5B). Using this synthetic exon-based approach, we aim to precisely engineer the IleE11 to ValE11 substitution in the Hbaa1.2, enhancing the efficiency of positive line identification and successful genome modification.

We co-injected the donor minicircle, sgRNA, and Cas9 and evaluated genome editing with $F_0$. The rate of *α-cry:mCherry*$^+$ $F_0$ larvae is 45.31% (29/64). We sorted *α-cry:mCherry*$^+$ larvae and performed genotyping to assess recombination events. Of these, 77% *α-cry:mCherry*$^+$ larvae (17/22) showed amplification of the 3′ junction, but 22% (5/22) had successful integration of the modified *hbaa1.2* coding DNA at both 5′ and 3′ junctions (Fig. S5B). The higher rate of 3′-only recombinants suggests unintended recombination between the 285 bp 3′ UTR and 3′ HA,

rather than 5′ and 3′ HA. We raised *α-cry:mCherry*$^+$ $F_0$ embryos to adulthood and screened their progeny for *α-cry:mCherry* expression. From approximately 40 $F_0$ individuals, we identified six founders. Of these, two exhibited incomplete KI, likely due to recombination between the 3′ UTR-pA and the 3′ HA, resulting in failure to integrate the 5′ synthetic exon (Fig. 5C). One founder carried the fully and accurately edited genome with correct integration at *hbaa1.2* locus (Fig. 5C,D). PCR analysis with primers specific to wild-type and edited *hbaa1.2* (*hbaa1.2*$^{ItoV}$) revealed that wild-type primers amplified *hbaa1.2* fragment from genomic DNA extracted of wild-type sibling but failed to amplify the fragment from *hbaa1.2*$^{ItoV}$ homozygotes. Conversely, *hbaa1.2*$^{ItoV}$ specific primers amplified the target fragment only in the *hbaa1.2*$^{ItoV}$ homozygotes (Fig. 5D). These results demonstrate that our strategy can achieve precise genome editing even in the presence of highly similar endogenous sequences.

To determine whether the edited *hbaa1.2*$^{ItoV}$ allele produces a protein without disrupting expression, we collected blood from adult *hbaa1.2*$^{ItoV}$ homozygous fish and performed mass spectrometry analysis. We successfully detected the peptides carrying the I-to-V substitution (Fig. 5E). Because *hbaa1.1* remains unmodified in *hbaa1.2*$^{ItoV}$ homozygotes, native I-containing peptides were also detectable. Notably, quantitative analysis revealed a higher abundance of V-containing peptides compared to I-containing ones, indicating that the edited *hbaa1.2*$^{ItoV}$ allele is actively expressed (Fig. 5F). These results suggest that our strategy for single a.a. substitution can serve as an effective genome editing approach.

## DISCUSSION

CRISPR/Cas9-mediated KIs rely on inducing a targeted DNA double-strand break (DSB) and providing a donor template for integration. Assembling donor templates – by combining two HAs with a gene of interest into a single plasmid – requires heavy subcloning. To simplify labor-intensive cloning steps, we developed a mini-golden-mediated subcloning strategy, which enables rapid and efficient construction of donor templates. A diverse set of middle entry vectors supports the construction of donor templates carrying various genes of interest, including fluorescence protein with subcellular organelle targeting sequences, the *CreER* recombinase,

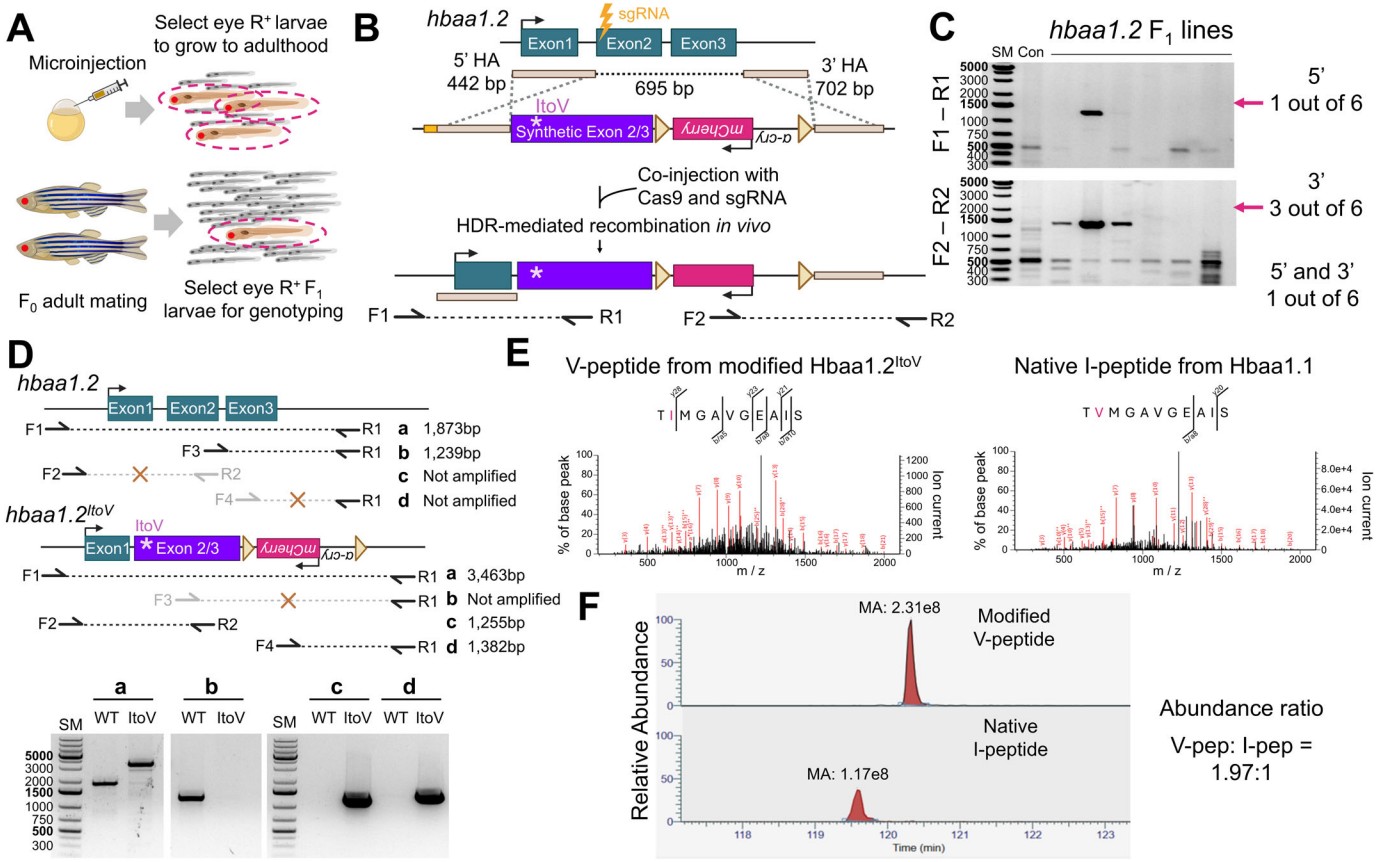

**Fig. 5. Synthetic exon-mediated genome editing combined with fluorescence-based screening successfully engineers the single amino acid of Hbaa1.2.** (A) Fluorescence-based screening enriches for larvae carrying the edited genome, facilitating efficient identification of correctly modified individuals. (B) Schematic of genome editing strategy to substitute I with V in Hbaa1.2. (C) PCR genotyping of $F_1$ stable lines targeting the upstream (F1-R1) and downstream (F2-R2) regions flanking the integration site at the *hbaa1.2* locus. (D) PCR analysis comparing wild-type (WT) siblings and *hbaa1.2^{ItoV}* homozygotes (ItoV) to distinguish WT and *hbaa1.2^{ItoV}* alleles. (E) Mass spectrometry analysis of blood from *hbaa1.2^{ItoV}* homozygote fish. The edited peptide containing I to V substitution (V-pep) is detectable. (F) Relative abundance of the edited V-pep versus the native I-pep. Panels A, B, D were created in BioRender by Kang, J., 2026. https://BioRender.com/6ix3ujm, https://BioRender.com/frw99cn and https://BioRender.com/rur146f. This figure was sublicensed under CC-BY 4.0 terms.

the *NTR* gene for genetic ablation, and more. In addition, Golden Gate subcloning allows the flexible incorporation of auxiliary DNA modules between the middle entry fragment and either the 5′ or 3′ HA, offering enhanced flexibility in construct design. Middle entry fragments can also be assembled from multiple modular components – ideal for generating fusion proteins – thus expanding the versatility of this system. Our strategy requires to mutate BsaI sites within HAs for the Golden Gate assembly. Although the impact of sequence mismatches on HDR-mediated KI is not fully understood, we successfully created the *foxd3^{CreER}* KI line despite the presence of a BsaI mutation in the 5′ HA. Together with recent studies showing that short HA (40-50 bp) can support efficient HDR-mediated KI (Wierson et al., 2020; Mi and Andersson, 2023; Oikemus et al., 2025), our results suggest that HDR can tolerate limited sequence mismatches within the longer HAs without substantially compromising KI efficiency. To ensure compatibility with the established KI methods such as GeneWeld (Wierson et al., 2020), we also provide destination vectors carrying a universal sgRNA site at the 5′ end. Furthermore, the middle entry library can serve as a template for generating short-arm donor fragments containing 5′ modifications, such as C6 linker (AmC6) (Mi and Andersson, 2023), 2′O-methyl-end modification (2′ OMe) (Oikemus et al., 2025), biotin (Gutierrez-Triana et al., 2018; Zhang et al., 2023) or other chemical modification. Altogether, we present the mini-golden system as a flexible and efficient toolkit for CRISPR-based genome editing applications.

Early strategies for introducing small edits, such as single nucleotide variant or short DNA insertion, utilized single-stranded oligodeoxynucleotides (ssODNs) containing both HAs and variant sequence (Burg et al., 2018; Prykhozhij et al., 2018; Tessadori et al., 2018). More recent advances in genome editing include prime and base editing approaches. Prime editing employs a Cas9 nickase fused to a reverse transcriptase and a prime editing guide RNA (pegRNA) to directly write new sequences into the genome without donor templates (Petri et al., 2022). Similarly, base editing (using cytidine or adenine deaminase fused to Cas9) has been applied in zebrafish to precisely convert single bases without inducing DSB (Rosello et al., 2021). In particular, the recent advancement of base editing techniques has robustly improved efficiency (Zheng et al., 2023; Qin et al., 2024; Liu et al., 2025; Oikemus et al., 2025; Zhong et al., 2025). However, these approaches still have some limitations, including unintended conversion of bystander nucleotides and a limited set of validated conversion options. To date, C-to-T and A-to-G edits have been shown to be effective in zebrafish, whereas efficient conversion of other types has not yet been confirmed (Liu et al., 2025). Another challenge is how to effectively identify animals carrying the edited genome. Our strategies address this challenge in two important ways. First, our strategy enables convenient and robust enrichment of larvae harboring edited genome using fluorescence signal. Our method allows reliable identification of edited animals simply by monitoring fluorescent reporter expressions, thereby

focusing downstream analysis on animals with the edits. This also improves genotyping sensitivity by avoiding dilution of edited gDNA with that of unedited animals.

Second, our synthetic exon approach minimizes undesired recombination events. Previous studies to insert two *loxp* sites also used fluorescent screening enrichment (Shin et al., 2023). However, those approaches placed wild-type sequences between two HAs, resulting in low efficiency due to unwanted recombination between either HA or the homologous endogenous target exon. Our synthetic exon-mediated strategy can mitigate this by avoiding direct sequence homology with the endogenous exon. Nevertheless, codon replacement within the synthetic exon may lead to unexpected outcomes, such as affecting mRNA stability (Wu and Bazzini, 2023). To minimize such effect, optimal synonymous codon substitutions should preserve the native codon usage balance. To support this, we provide practical guidelines and templates (Fig. 6 and supplementary material). While our edited *hbaa1.2* allele is expressed, further validation with other gene targets will be necessary to fully optimize the synthetic exon-mediated genome editing approach.

Overall, our strategy offers an efficient method to generate KI lines, expanding the capacity to create KI animals and enabling deeper investigation into gene function across diverse biological processes.

## MATERIALS AND METHODS
### Zebrafish
Wild-type or transgenic male and female zebrafish of the outbred Ekkwill (EK) or AB strain ranging up to 18 months of age were used for all zebrafish experiments. The following transgenic lines were used in this study: *Gt(foxd3:mCherry)^ct110aR^* (Hochgreb-Hagele and Bronner, 2013), *Tg(–3.5ubi:loxP-EGFP-loxP-mCherry)^cz1701Tg^* or *ubi:switch* (Mosimann et al., 2011), *foxd3^CreER#9^* (*uwk45*), *foxd3^CreER#23^* (*uwk46*), and *hbaa1.2^ItoV^* (*uwk47*). The water temperature for adult animals was maintained at 26°C unless otherwise indicated. Embryos and larvae were maintained at 28°C in egg water containing 300 mg/L sea salt, 75 mg/L calcium sulfate, 37.5 mg/L sodium bicarbonate, and 0.0001% Methylene Blue. Animals were anesthetized in 0.02% tricaine until gill movement stopped. *flp* mRNA was generated using the pCS2-Flp plasmid and the mMessenger mMachine SP6 kit (Thermo Fisher Scientific). For 4-hydroxytamoxifen (4-HT) treatments, zebrafish larvae were incubated with 5 uM of 4-HT (Sigma-Aldrich, H7904) for three

consecutive days from 5 to 77 hpf with daily media replacement. Work with zebrafish was performed in accordance with University of Wisconsin-Madison guidelines.

RNA was isolated from larvae using Tri-Reagent (Thermo Fisher Scientific). Complementary DNA (cDNA) was synthesized from 300 ng to 1 μg of total RNA using a NEB ProtoScript II first strand cDNA synthesis kit (NEB, E6560). Primer sequences used for RT-PCR are listed in Data File 1.

### Subcloning and transgenesis
mini-golden destination vector: To generate the pMC-Dest-BsaI destination vector (pMC-Dest-BsaI; Addgene ID: 200550), three BsaI sites in the pMC.BESPX-MCS2 (MN100B-1, SBI) were mutated using a site-directed mutagenesis kit (NEB, E0554S). A double-stranded oligonucleotide that contains two BsaI sites producing CTCA and CCAA overhang at the 5′ and 3′, respectively, was inserted via EcoRI and SalI enzymes-mediated subcloning. A recent study reported that addition of a reporter cassette outside of the HAs can aid in detecting off-target insertions (Shin et al., 2023). To incorporate this strategy, we further modified the destination vector by inserting a *cmlc2:mCherry-pA* cassette via PspOMI and SalI enzymes-mediated subcloning (pMC-Dest-BsaI-cmlc2:mCherry; Addgene ID: 241213). To expand the mini-golden system for compatibility with the GeneWeld method (Wierson et al., 2020), we also added a universal sgRNA sequence (GGGAG GCGTT CGGGC CACAG CGG) at the 5′ end of the cloning site via PspOMI and SalI enzymes-mediated subcloning (pMC-Dest-uni-gRNA5-BsaI; Addgene ID: 241150). Primers used for subcloning are listed in Data File 1.

Middle entry library vector version 1: The middle entry vector backbone was derived from Addgene plasmid #64247 (pU6a:sgRNA#3) (Yin et al., 2015), which carries a spectinomycin resistance gene. pU6b:sgRNA#3 was a gift from Wenbiao Chen (Addgene plasmid #64247; http://n2t.net/addgene:64247; RRID:Addgene_64247). Two new BsaI sites generating AGTC and CCAA overhangs at the 5′ and 3′ end, respectively, replaced the *pU6a* promoter sequence via PspOMI and XbaI enzymes-mediated subcloning. Insert fragments were prepared by PCR with primers tagged with BsaI site to produce AGTC and CCAA overhang at the 5′ and 3′ end, respectively. After digestion of the vector and the insert with BsaI, linearized fragments were purified using a gel extraction kit (Takara Bio, 740609.250), ligated with T4 ligase (NEB, M0202 M), and transformed into *E. coli* for further subcloning. An "AGTC" linker sequence remains between the 5′ HA and a gene of interest.

Middle entry library vector version 2: We are aware that "AGTC" could potentially act as a splicing acceptor (AG) or donor (GT) site, possibly leading to unintended splicing events when integrated into the genome. To

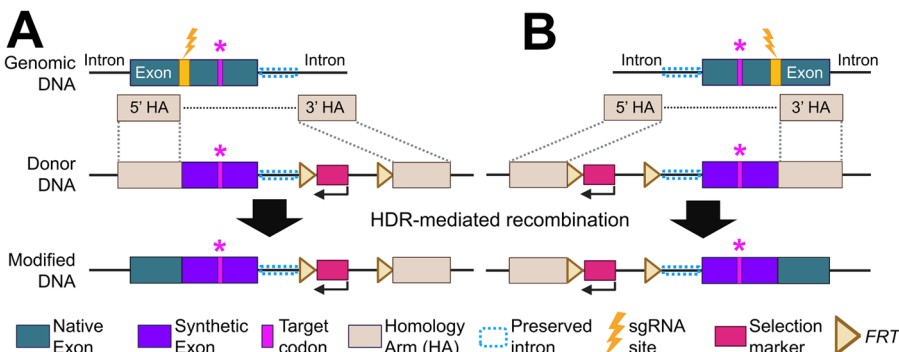

**Fig. 6. The schematic of the proposed design for single amino acid substitution.** A recent study reported that placing HAs more than 5 bp from the cut site reduces recombination efficiency (Oikemus et al., 2025). Our data demonstrates that recombination can occur efficiently even when two HAs are positioned distantly. Taken together, these findings suggest that at least one HA may need to be placed within 5 bp of the cut site to maintain high efficiency. Based on this framework, we propose two potential design strategies. (A) If the sgRNA site is located upstream of the target codon, the 5′ HA ends at the cut site and the *FRT*-franked fluorescence selection marker can be inserted within the following intron in the reverse-complement orientation. (B) If the sgRNA site is located downstream of the target codon, the 3′ HA begins at the cut site, and the *FRT*-franked fluorescence selection marker can be inserted into the preceding intron in the reverse-complement orientation. In both cases (A,B), codons between the sgRNA cut site and the end of exon, excluding the target codon, will be replaced with synonymous substitutions. Because exon-intron junctions contain key regulatory elements, we recommend preserving 30-40 bp of the native intronic sequence to avoid disrupting splicing.

address this concern, we resubcloned middle entry vectors using TCGC and CTCA overhangs at 5′ and 3′ end, respectively. Plasmids are listed in Table S1. Primers used for subcloning are listed in Data File 1.

*foxd3 CreER* donor construct: 5′ and 3′ HAs were amplified using genomic DNA extracted from fish that were used for injection. The corresponding sgRNA site for *foxd3* was positioned at the 5′ end of the 5′ HA. The sgRNA and PAM sequences for *foxd3* are "GCACA GGTGA GCGAC GCATG TGG". These HA fragments and pMC-ME_*CreER-pA-FRT-acry:mCherry-pA-FRT* were assembled with pDest-BsaI via BsaI-mediated golden gate reaction. To purify the minicircle vectors, parental plasmids were transformed into the ZYCY10P3S2T *E. coli* Minicircle producer strain (MN900A-1, SBI). Minicircles were prepared as described in Keating et al. (2024). Primers used for subcloning are listed in Data File 1.

*hbaa1.2* modified donor construct: The sgRNA and PAM sequences for *hbaa1.2* are "GGTCT TGGTC TGAGG ATAGA CGG". The fragment containing sgRNA, 5′ HA, and modified exon sequence was synthesized as gBlocks by IDT and subcloned into pMC.BESPX-MCS2 via EcoRI and ClaI enzymes-mediated subcloning. The 3′ HA was subsequently amplified from genomic DNA extracted from the fish used for injection and inserted into the vector via BamHI and SacI enzyme-mediated subcloning.

Finally, the FRT-flanked α-*cry:mCherry-pA* fragment was inserted in the reverse complement orientation using BamHI and ClaI enzyme-mediated subcloning. To purify the minicircle vectors, parental plasmids were transformed into the ZYCY10P3S2T *E. coli* Minicircle producer strain (MN900A-1, SBI). Minicircles were prepared as described in (Keating et al., 2024). Primers used for subcloning are listed in Data File 1.

### Generation of *foxd3^CreER^* and *hbaa1.2^ItoV^* KI lines and genotyping

sgRNAs were synthesized by a cloning-free method as described in Varshney et al. (2015). Briefly, primers containing T7 promoter, sgRNA target and partial sgRNA scaffold sequences were annealed with the universal sgRNA 3′ scaffold primer, and the annealed oligos were filled using the thermocycler and PCRBio polymerase (Genesee Scientific) to generate sgRNA templates. sgRNAs were synthesized using the HiScribe T7 kit (NEB, E2050S) and purified by the RNA purification Kit (Zymogen, R1016) according to the manufacturer's instructions. A sgRNA (25-30 ng/ul) and a donor minicircle (20-25 ng/ul) were mixed and co-injected with Cas9 protein (0.5 μg/ul; PNABio, CP01) into the one-cell stage embryos. Genomic DNA was extracted, and KI was confirmed by genotyping. Primers used for genotyping are listed in Data File 1. Larvae having mCherry in lens were sorted and raised to adulthood, and founders were screened with $F_1$ progenies. The founder was outcrossed with wild-type animals to generate heterozygous animals.

### Alcian Blue cartilage staining

To assess cartilage development, we used an acid-free Alcian Blue staining method (Balasubramanian et al., 2023) on 5 dpf larvae. Larvae were first washed in PBS and then euthanized by cold-shock on ice for 10 min. Larvae were transferred to 1.5 ml microcentrifuge tubes containing 4% paraformaldehyde (PFA; VWR, 102091-904) and incubated for two nights at 4°C. Following fixation, samples were rinsed twice in PBST (PBS+0.1% Tween-20) for 5 min each. Larvae were dehydrated in 50% ethanol and rocked at room temperature for 10 min. Ethanol was replaced with 1 ml of Alcian Blue staining solution (final concentrations of: 0.1 mg/ml Alcian Blue in methanol, 60 mM MgCl2, and 70% ethanol). Samples were rocked overnight at room temperature and protected from light using foil. Stained larvae were rinsed once in PBST then transferred to a 24-well culture plate containing 2 ml bleaching solution (final concentrations of: 3% $H_2O_2$ and 0.5% KOH). Samples were incubated for 30 min in the dark. Following bleaching, larvae were rinsed again in PBST. Larvae were cleared by rocking in 25% glycerol with 0.25% KOH for 1 h, followed by incubation overnight in 50% glycerol with 0.25% KOH. Larvae were stored in 50% glycerol with 0.1% KOH at 4°C.

### Imaging

Whole-mount larval images were acquired using an AxioZoom stereo fluorescence microscope (Zeiss) or an Olympus FV3000 confocal microscope (Olympus). Whole-mount imaging for Alcian Blue stained larvae was acquired using an AxioZoom and color camera. Further image processing was carried out manually using Zen (Zeiss), Fluoview (Olympus), Photoshop, or FIJI/ImageJ software.

### Blood collection and mass spectrometry

Our blood collection method was adapted from Babaei et al. (2013). A 22G needle was used to bore a single hole in the bottom of a 0.6 ml centrifuge tube, followed by four additional equidistant holes made with a 27G needle around the central hole. This modified 0.6 ml tube was then placed into a 1.5 ml centrifuge tube. A 15 μL drop of 20 U/ml heparin in PBS was added to the side of the inner 0.6 ml tube. Fish weighing more than 0.4 g were anesthetized with 0.02% tricaine (MS222), then gently dried with a paper towel to remove excess water. Using a straight blade dipped in heparin solution, the fish tails were amputated just above the base of the caudal fin. The fish were quickly transferred cut-side down, into the 0.6 ml tube, ensuring that the cut tail contacted the heparin droplet. After closing the lid of the inner tube, samples were centrifuged at 60× *g* for 5 min at 14°C. The collected blood was gently flicked to mix with the heparin solution to minimize clotting. Samples were kept on ice until used in downstream applications. Fish carcasses were subsequently used for genomic DNA (gDNA) and RNA extraction.

Hb peptides were prepared using trypsin/LysC and iodoacetic acid (Kassa et al., 2021). Dithiothreitol final concentration was 2 mM. Samples were desalted with a C18 OMIX tip. Samples were injected into a Pepmap C18, 3 μM, 100A, 75 μM ID, 15 cm reversed phase column. Samples were analyzed on an Orbitrap Elite coupled to an EASY-Spray ion source in data dependent MS/MS mode.

### Protocol: mini-golden system to assemble a donor construct for generating KI lines

1. Choose your KI strategy
   a) Targeting upstream of ATG [promoter region or 5′ untranslated region (UTR)].
      - Pros: avoids concerns about out-of-frame editing.
      - Con: if multiple transcription start sites exist, the resulting KI may not recapitulate expression of all isoforms.
   b) Targeting within an exon or 3′ end
      - Pros: enables faithful recapitulation of endogenous expression.
      - Con: expression depends on in-frame integration; out-of-frame edits will not result in functional reporter expression.
2. Select sgRNA site.
Two options are available.
   a) sgRNA site located within the 5′ HA
      - The 5′ HA typically begins at the sgRNA target site.
      - After editing, the sgRNA site in the genome is often mutated, preventing re-cutting.
   b) sgRNA site is located between 5′ and 3′ HAs
      - The edited genome will not retain the sgRNA site, avoiding re-cleavage.
      - The optimal distance between the 5′ and 3′ HAs for efficient editing remains uncertain. In our experience, HDR-mediated recombination can occur with up to a 700 bp interval between the two HAs.
      - The optimal distance between cutting site and HAs remains uncertain. Based on recent study (Oikemus et al., 2025), longer than 5 bps may result in poor outcome. Thus, it is recommended to place at least one of HAs close to the cutting site.
Note: efficient sgRNA can be predicted via CRISPRscan (www.crisprscan.org/) (Moreno-Mateos et al., 2015).
3. Choose middle entry vector
Select an appropriate middle entry vector based on your experimental needs:
   - Vectors containing 2A sequences:
     pMC-ME plasmids with *EGFP*, *mCherry*, or *BFP* may lack the GSG linker sequence upstream of the 2A peptide. Please check the vector map.
   - Vectors containing splicing acceptors (SA):
     These vectors mimic enhancer or gene trap strategies. SA sequence is derived from (Ichino et al., 2020).

Biology Open

4. Design primers with BsaI enzyme sites

Design primers to amplify 5′ and 3′ HAs with appropriate overhangs and BsaI recognition sites. If a BsaI site exists within your HA, it must be mutated. In-fusion PCR can be used for this purpose.

Note: the sgRNA site in the minicircle will be cleaved in the cell, generating a linearized donor construct.

### Primer structure

5′ HA CAGT forward primer: atc ggtctc t CAGT (sgRNA sequence of target region+PAM) (target region 5′ forward sequence)

5′ HA TCGC reverse primer: gat GGTCTC T GCGA (target region 5′ reverse sequence)

3′ HA CTCA forward primer: atc GGTCTC T CTCA (target region 3′ forward sequence)

3′ HA CCAA reverse primer: gat GGTCTC T TTGG (target region 3′ reverse sequence)

5. Amplify homology arm fragments

Run PCR using primers and genomic DNA extracted from fish that will be used for injection. Purify PCR products for 5′ and 3′ HA fragments via gel extraction approach.

6. Perform Golden-Gate reaction

Set up the Golden-Gate ligation reaction with the following components:

- 1 µl T4 DNA ligase buffer
- 25-50 ng each of 5′ and 3′ HA cassettes
- 100 ng chosen middle entry (pMC-ME) vector
- 100 ng destination vector
- 0.5 µl BsaI-HF v2
- 0.5 µl T4 DNA ligase
- Add nuclease-free water to a final volume of 10 µl

7. Incubate mixture 6× (37°C for 20 min, 16°C for 15 min), followed by 37°C for 30 min and 80°C for 15 min.

8. The reaction is ready for transformation (use 5 µl of the ligation and plate 20% of the transformants). Transform and spread onto kanamycin (100 µg/ml) plates.

9. Screen positive colonies by colony PCR.

10. Purify a plasmid and confirm it by enzyme mapping or sequencing.

11. Perform minicircle DNA purification and inject into the one-cell stage zebrafish embryos. See the details in Keating et al. (2024).

12. Final vector sequence order.

Version 1:

vector - CAGT - sgRNA – 5′ HA – AGTC – pME – CTCA – 3′ HA – CCAA – vector.

Version 2:

vector - CAGT - sgRNA – 5′ HA – TGCG – pME – CTCA – 3′ HA – CCAA – vector.

### Protocol: mini-golden system to assemble a donor construct for single amino acid change

1. Select an sgRNA near the target amino acid

Note: efficient sgRNA can be predicted via CRISPRscan (www.crisprscan.org/) (Moreno-Mateos et al., 2015).

2. Validate sgRNA cutting efficiency

Inject sgRNA and Cas9 into embryos, followed by an sgRNA efficiency assay (e.g. T7 endonuclease assay) to assess cleavage at the target site.

3. Modify codons using synonymous substitutions.

Modify codons from the target amino acid (or from the sgRNA site) to the end of the same exon using synonymous codons. Codon replacement may lead to unexpected outcomes, such as affecting mRNA stability (Wu and Bazzini, 2023). Therefore, minimizing replacement is important. To address this, we follow these guidelines:

- Swapping: replace codons with synonymous alternatives that maintain the overall codon usage balance. For example, glycine (G) can be encoded by GGT, GGC, GGA, or GGG. If the target exon contains two instances each of GGC and GGG, we may swap GGC with GGG to preserve codon composition.
- Altering: when swapping is impractical, we substitute with the most frequently used synonymous codon based on the zebrafish codon usage table (Subramanian et al., 2022).

- Unchanged: codons encoding methionine (M) and tryptophan (W), which each have only a single codon, will remain unaltered.

4. Complete the template (File 13) and design a synthetic exon that minimizes codon altering but prioritizes codon swapping where possible.

5. Choose 5′ and 3′ HA region.

Note: The Kang lab typically uses 400-500 bp fragments for each HA.

6. The synthetic fragment information: contain sgRNA, 5′ HA, and a synthetic exon.

### Fragment structure

atc ggtctct (BsaI site) – CAGT – sgRNA+PAM – 5′ HA – altered target amino acid – codon-substituted region – TCGC agagacc (BsaI site) atc

7. To detect whether synthetic exon sequences generate *de novo* splicing donor or acceptor site, it is recommended to review the sequence using software, such as spliceator (www.lbgi.fr/spliceator/) (Scalzitti et al., 2021). If score is high, change to other synonymous alternatives.

8. Perform *in silico* subcloning

Simulate the cloning strategy to ensure proper assembly and sequence integrity.

Vector - CAGT - sgRNA – 5′ HA – altered amino acid – a codon substituted region TCGC – pMid_acrymCherry-FRT – CTCA – 3′ HA – CCAA – vector

9. Synthesize a synthetic exon (e.g. IDT gBlock option) and order primers from the company

### Gene fragment for synthesis

atc ggtctct (BsaI site) - CAGT - sgRNA – 5′ HA – Altered amino acid – a codon substituted region – TCGC agagacc (BsaI site) atc.

### Primer structure

5′ HA CAGT forward primer: atc ggtctc t CAGT (sgRNA sequence of target region) (target region 5′ forward sequence)

5′ HA TCGC reverse primer: gat GGTCTC T GCGA (target region 5′ reverse complement sequence)

3′ HA CTCA forward primer: atc GGTCTC T CTCA (target region 3′ forward sequence)

3′ HA CCAA reverse primer: gat GGTCTC T TTGG (target region 3′ reverse complement sequence)

10. Perform PCR to get 5′HA-synthetic-exon and 3′ HA and assemble all fragments into pDest vector via Golden gate reaction.

- 1 µl T4 DNA ligase buffer
- 25-50 ng each of 5′ HA-synthetic exon fragment and 3′ HA cassette
- 100 ng pMC-ME-v2_FRT-acrymCherry-FRT
- 100 ng destination vector
- 0.5 µl BsaI-HF v2
- 0.5 µl T4 DNA ligase
- Add nuclease-free water to a final volume of 10 µl

Note: 5′ HA-synthetic exon can be obtained by PCR using synthetized fragment as a template.

Note: 3′ HA can be obtained by PCR using gDNA as a template.

Incubate mixture 8× (37°C for 20 min, 16°C for 15 min), followed by 37°C for 30 min and 80°C for 15 min.

11. The reaction is ready for transformation (use 5 µl of the ligation and plate 20% of the transformants). Transform and spread onto kanamycin (100 µg/ml) plates.

12. Screen positive colonies by colony PCR.

13. Purify a plasmid and confirm it by enzyme mapping or sequencing.

14. Perform minicircle DNA purification and inject into the one-cell stage zebrafish embryos. See the details in Keating et al. (2024).

### Acknowledgements

We thank the UW–Madison SMPH BRMS staff and members of the Kang Lab for zebrafish care; Sarah C. Kucenas and Marianne Bronner for *Gt(foxd3: mCherry)^{ct110aR}*; Len Zon and Jingli Cao for *ubi:switch*; Rachel Wong and Owen Lawrence for sharing a membrane localization sequence containing plasmid;

Addgene, Wenbiao Chen, and depositors for sharing plasmids; Biorender for image creation; and Mr. Greg Sabat for mass spectrometry analysis and interpretation.

**Competing interests**
The authors declare no competing or financial interests.

**Author contributions**
Conceptualization: A.R.-P., J.K.; Data curation: A.R.-P., E.G.B., S.M.; Formal analysis: A.R.-P.; Funding acquisition: M.P.R., J.K.; Investigation: A.R.-P., S.M., I.S., S.I.A., S.B., J.K.; Methodology: A.R.-P., E.G.B., S.M., I.S., S.I.A., M.P.R., S.B., J.K.; Project administration: J.K.; Resources: S.C., H.A., M.G.; Supervision: M.P.R., J.K.; Validation: A.R.-P.; Visualization: A.R.-P.; Writing – original draft: J.K.; Writing – review & editing: A.R.-P., J.K.

**Funding**
This work was supported by National Institutes of Health (R35GM137878, R01HL151522, R21OD037634 and P30CA014520 to J.K.); Vilas Faculty Early-Career (GR000042507 to J.K.); Investigator Award; National Institute of Food and Agriculture, USDA Hatch project (7000320 to M.P.R.); Improving Food Quality Foundational Program (2019-67017-29179 to M.P.R.); and National Science Foundation Graduate Research Fellowship (2137434 to M.G.). Open Access funding provided by NIH. Deposited in PMC for immediate release.

**Data and resource availability**
All relevant data and details of resources can be found within the article and its supplementary information.

**Peer review history**
The peer review history is available online at https://journals.biologists.com/bio/lookup/doi/10.1242/bio.062472.reviewer-comments.pdf

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
