## [Peer Review File · Biology Open]

Advancing Knock-In Approaches for Robust Genome Editing in Zebrafish

Anjelica Rodriguez-Parks, Ella Grace Beezley, Steffani Manna, Isabella Silaban, Sarah Almutawa, Siyang Cao, Hossam Ahmed, Megan Guyer, Mark Richards, Sean Baker and Junsu Kang

DOI: 10.1242/bio.062472

Editor: Daniel Gorelick

Review timeline

Submission to sister journal:	15 August 2025
Editorial decision at sister journal:	7 October 2025
Transfer to Biology Open:	18 December 2025
Editorial decision:	23 December 2025
Resubmission received:	7 January 2026
Accepted:	14 January 2026

Original submission to sister journal

First decision letter

MS Title: Advancing Knock-In Approaches for Robust Genome Editing in Zebrafish

Authors: Anjelica Rodriguez-Parks, Ella Grace Beezley, Steffani Manna, Isabella Silaban, Sarah Almutawa, Siyang Cao, Hossam Ahmed, Megan Guyer, Mark Richards, Sean Baker and Junsu Kang

I have now received all the referees' reports on the above manuscript, and have reached a decision. I am sorry to say that the outcome is not a positive one. The referees' comments are appended below, or you can access them online: please go to.

As you will see, the referees raise some significant concerns about your paper, and are not strongly in favour of publication. In reviews and in post-review discussion, the reviewers consider that there is insufficient methodological advance and conceptual novelty to justify publication in this journal. Given their opinions, I must therefore, reject your paper.

I do realise this is disappointing news, but this journal receives many more papers than we can publish, and we can only accept manuscripts that receive strong support from referees.

I do hope you find the comments of the referees helpful, and that this decision will not dissuade you from considering our journal for publication of your future work. Many thanks for sending your manuscript to us.

Reviewer 1

Comments for the author

The manuscript by Rodriguez-Parks et al., describes a set of GateWay vectors for generating plasmids with long homology arm-flanked templates for isolating knock-ins in zebrafish. The title, Efficient Knock-in Approaches for High-Precision Genome Editing in Zebrafish, does not represent the major findings of the study. The manuscript describes the application of an established knock-in method for targeted integration in zebrafish that uses homology arms to drive integration via Homology Directed Repair - as previously reported and validated in numerous publications. Vectors

© 2026. Published by The Company of Biologists under the terms of the Creative Commons Attribution License (<https://creativecommons.org/licenses/by/4.0/>).

in the present study are used to generate mini-circles, to prevent incorporation of the vector backbone, as described previously by this group in Keating et al., 2024. A set of GateWay cloning vectors is described for assembling targeting constructs containing fluorescent reporters, CreER, NTRv2, or other cDNAs.

The paper does not describe an advance to existing methods for generating precision KI alleles of long DNA segments by HDR in zebrafish. Some vectors contain a linked secondary marker to facilitate screening for injected embryos and stable transgenic knock-in lines, as previously described in work from others - Almeida et al., 2021; Liu et al., 2022; Ming et al., 2025; etc. As written, the study claims to recover a hbaa1.2-Isoleucine >Valine line by "engineering of a single amino acid substitution" in the hbaa1.2 gene, providing a major advance for SNP integration/disease modeling. Instead, a cassette containing the remaining cDNA of the gene with the substitution mutation was integrated into the coding sequence of exon 2.

In the current study, two knock-in lines were described.

The first knock-in line:

A single gRNA site located in the 5' UTR was used to target a creER, a-cry:mCherry into foxd3 cassette using 500bp plus homology arm sequences that mapped downstream of the gRNA cut site within the foxd3 gene. This approach is confusing, and should be clearly described as a deletion plus replacement integration. Here the integration is a deletion of the sequences in between the two homology arms and replacement with the cre cassette. 1/7 transmitted what is described as a precise integration, similar to published frequencies using GeneWeld short homology directed repair to precisely integrate cassettes at the gRNA cut site (Wierson et al., 2020; Almeida et al., 2021; Ming et al., 2025).

The line results in a foxd3 deletion loss of function, or complex duplication/rearrangement. The advantage of this approach over recently described methods for efficient recovery of endogenous cre and creERT2 precise integrations, without deletion or rearrangement, at the 3' end of the coding sequence of a gene of interest, either by linear 5' blocked templates (Mi and Andersson, 2023; Zhang et al., 2023; others) or plasmid-based GeneWeld (Ming et al., 2025) is not explained.

The second knock-in line:

The heading for this part of the manuscript inaccurately claims "engineering of a single amino acid substitution in hbaa1.2", implying recovery of precise substitution mutation at the endogenous gene codon. A cDNA cassette containing a substitution mutation and linked secondary marker was integrated into exon 2 of the hbaa1.2 gene to express an Hbaa1.2-I>V variant protein. This approach integrated a cDNA cassette into exon 2 (in frame? Details were not presented) to express the remainder of the exon2-3 coding with the substitution mutation of interest. This approach was incorrectly described as using a "synthetic exon" - the cassette contained an exon2-3 cDNA (with endogenous exon 3 3' UTR/transcriptional terminator? This was not validated). Synthetic exons contain at least a splice acceptor and meet the functional definition of an exon. 1/40 founders transmitted a precise integration allele, which is not unreasonable.

The authors went to considerable effort to demonstrate the hbaa1.2 I>V cDNA line does express Hbaa1.2I>V peptide. A protocol was provided for vector assembly.

Reviewer 2: SUMMARY OF THE ADVANCE MADE IN THIS PAPER AND ITS POTENTIAL SIGNIFICANCE TO THE FIELD

This manuscript describes the development of the mini-golden Golden Gate-based donor construction platform for zebrafish knock-in strategies and a synthetic exon-based template approach to improve identification of correctly edited alleles. The authors demonstrate its use to generate a foxd3CreER KI line and to engineer a single amino acid substitution in hbaa1.2. In principle, the work provides an additional toolkit for streamlining donor vector construction and improving precision in CRISPR-mediated knock-ins. The potential significance lies in easing cloning steps and providing fluorescence-based selection for editing, though the overall biological novelty is limited. In addition the use of synthetic exons for single bp substitution is not always a desirable option as changing the overall exon-intron gene organization can affect transcription. How can this be applied to very long genes when the mutation is in the first exons of 10 or more? Would the authors suggest to eliminate entire kilobases of intron sequences?

SUGGESTIONS TO AUTHORS

The mini-golden system is essentially a repackaging of Golden Gate cloning for donor assembly, and its added value over existing modular strategies needs to be more convincingly demonstrated (for instance Gibson cloning). Direct quantitative comparisons to GeneWeld, cloning free approaches such as the ones proposed by Almeida et al, prime/base editing, or other recent methods are required.

The method is tested at only two loci, which is insufficient to support broad claims of generality. Additional targets should be included to provide statistical support for the efficiency and reproducibility of the approach.

Some claims in the introduction are misleading. For example, base editors are portrayed as low-efficiency tools, but multiple groups have demonstrated efficiencies approaching 100% in zebrafish and other systems. A comprehensive and up-to-date review of base editing (Liu et al., *Frontiers in Genome Editing* 2025) should be cited, and the discussion revised to more accurately reflect the state of the field. Given these advances, base editors remain the method of choice for single nucleotide substitutions when feasible.

Reviewer 3: SUMMARY OF THE ADVANCE MADE IN THIS PAPER AND ITS POTENTIAL SIGNIFICANCE TO THE FIELD

This manuscript details a GoldenGate cloning strategy for making knockins, including Cre reporter lines and substitutions in zebrafish. The work is well documented and will be a valuable resource for the broader community. There are no major developmental insights here, but the work would fall under a techniques and resources and would be acceptable for this journal.

SUGGESTIONS TO AUTHORS

The main thing I would like to see, in both examples, is a demonstration that the Cry-mCherry selection cassette can be cleanly removed with Flippase and this doesn't interfere with anything. This is most critical for assessing the Cre induced expression (Figure 3) because the Cry expression could be masking ectopic Cre expression. This is even more important because the mCherry in the KI line gets dysregulated and expressed more broadly than just the lens.

It would also be important to show this in the substitution line as well; I'd especially want to see that the protein levels with and without the Cry-mCherry remain similar. The impact of this selection cassette on expression will be of interest to groups thinking about applying this method.

Specific comments:

-Page 5: "Any internal BsaI sites in HAs must be mutated via overlap extension PCR or other available methods." What will that do to the efficiency of KI?

-The schematic in Fig 1A shows a sgRNA site and a cartoon of the primer for 5' and 3' amplification, but the logic of this is not made super clear here.

-Fig S1 C and D, labelling the ladder to easier understand the sizes would be helpful

-The description of the F1 generated fish in Fig 2B is confusing to follow because the text refers to fish ID numbers but only shows us a few fish, and the reader needs to keep remembering: 9= NHEJ, 23= HDR. Why not name the new alleles not with numbers but something more descriptive?

-In 2E and 2F, penetrance of the jaw defects is not large; how does that compare to homozygous LOF?

-Figure 3—the mCherry expression from the Cry promoter KI could be masking mis-expression of the Cre in the Ubi-switch. Test after first flipping out the Cry marker?

-Figure 4-5 The approach to use the Cry expression to screen fish for KI is interesting, but requires extensive modification of the exon-intron structure in addition to the substitution. I therefore wouldn't call it a precise substitution method, which to my mind is a change of specific bases only (maybe a few subs nearby but not artificial exons).

-In the expression analysis of the substitution, it would be useful to see with and without the lens marker (e.g., Flp out) to show that expression is not altered by the presence of this (or to document the extent to which is is a factor)

Transfer to Biology Open

Author response to reviewers' comments

Responses to Referees

Reviewer 1: The manuscript by Rodriguez-Parks et al., describes a set of GateWay vectors for generating plasmids with long homology arm-flanked templates for isolating knock-ins in zebrafish. The title, Efficient Knock-in Approaches for High-Precision Genome Editing in Zebrafish, does not represent the major findings of the study. The manuscript describes the application of an established knock-in method for targeted integration in zebrafish that uses homology arms to drive integration via Homology Directed Repair - as previously reported and validated in numerous publications. Vectors in the present study are used to generate mini-circles, to prevent incorporation of the vector backbone, as described previously by this group in Keating et al., 2024. A set of GateWay cloning vectors is described for assembling targeting constructs containing fluorescent reporters, CreER, NTRv2, or other cDNAs.

The paper does not describe an advance to existing methods for generating precision KI alleles of long DNA segments by HDR in zebrafish. Some vectors contain a linked secondary marker to facilitate screening for injected embryos and stable transgenic knock-in lines, as previously described in work from others - Almeida et al., 2021; Liu et al., 2022; Ming et al., 2025; etc. As written, the study claims to recover a hbaa1.2-Isoleucine >Valine line by "engineering of a single amino acid substitution" in the hbaa1.2 gene, providing a major advance for SNP integration/disease modeling. Instead, a cassette containing the remaining cDNA of the gene with the substitution mutation was integrated into the coding sequence of exon 2.

Reply: We appreciate the reviewer for the thoughtful comments. We also apologize for insufficient explanation to clarify our knock-in design and implication of our approach. In the revised manuscript, we have added new data clarifying our strategy and addressed the reviewer's concerns.

In the current study, two knock-in lines were described.

The first knock-in line:

A single gRNA site located in the 5' UTR was used to target a creER, a-cry:mCherry into foxd3 cassette using 500bp plus homology arm sequences that mapped downstream of the gRNA cut site within the foxd3 gene. This approach is confusing, and should be clearly described as a deletion plus replacement integration. Here the integration is a deletion of the sequences in between the two homology arms and replacement with the cre cassette. 1/7 transmitted what is described as a precise integration, similar to published frequencies using GeneWeld short homology directed repair to precisely integrate cassettes at the gRNA cut site (Wierson et al., 2020; Almeida et al., 2021; Ming et al., 2025).

The line results in a foxd3 deletion loss of function, or complex duplication/rearrangement. The advantage of this approach over recently described methods for efficient recovery of endogenous cre and creERT2 precise integrations, without deletion or rearrangement, at the 3' end of the coding sequence of a gene of interest, either by linear 5' blocked templates (Mi and Andersson, 2023; Zhang et al., 2023; others) or plasmid-based GeneWeld (Ming et al., 2025) is not explained.

Reply: We apologize for the confusion regarding our design for the foxd3^{CreER} line. Our goal was to

generate a *foxd3* loss-of-function allele as well as a *CreER* knock-in line by integrating *CreER* at the *foxd3* start codon and deleting a portion of *foxd3* coding sequences. In combination with existing *foxd3* mutants, this allele offers the advantage of creating a knockout through deletion of a relatively long genomic fragment, compared with short homology arm-based strategies, and it also serves as a genetic tool to study the roles of *foxd3*. We have clarified this design by revising the text.

We have added new text on Page 6:

We designed our strategy to integrate *CreER* into the 5' untranslated region (UTR) of *foxd3* and to delete a portion of the *foxd3* coding sequence to generate deletion mutants (Fig. 2A and Supplementary Fig. S1B).

The second knock-in line:

The heading for this part of the manuscript inaccurately claims "engineering of a single amino acid substitution in hbaa1.2", implying recovery of precise substitution mutation at the endogenous gene codon. A cDNA cassette containing a substitution mutation and linked secondary marker was integrated into exon 2 of the hbaa1.2 gene to express an Hbaa1.2-I>V variant protein. This approach integrated a cDNA cassette into exon 2 (in frame? Details were not presented) to express the remainder of the exon2-3 coding with the substitution mutation of interest. This approach was incorrectly described as a using a "synthetic exon" - the cassette contained an exon2-3 cDNA (with endogenous exon 3 3' UTR/transcriptional terminator? This was not validated). Synthetic exons contain at least a splice acceptor and meet the functional definition of an exon. 1/40 founders transmitted a precise integration allele, which is not unreasonable.

The authors went to considerable effort to demonstrate the hbaa1.2 I>V cDNA line does express Hbaa1.2I>V peptide. A protocol was provided for vector assembly.

Reply: We appreciate the reviewer for his/her overall support and critical comments on our manuscript. Nomenclature of synthetic exons that we use indicate coding sequences are changed. To clarify that our approach change codon, we have changed the title to indicate amino acid change rather than one single base pair change.

We have updated the title and the text on page 10:

Synthetic exon-based genome editing combined with fluorescence screening enables a single amino acid substitution in Hbaa1.2

A primary challenge in **genome editing for a.a. substitution** is the low efficiency of selecting animals that carry the desired modification.

Reviewer 2: SUMMARY OF THE ADVANCE MADE IN THIS PAPER AND ITS POTENTIAL SIGNIFICANCE TO THE FIELD

This manuscript describes the development of the mini-golden Golden Gate-based donor construction platform for zebrafish knock-in strategies and a synthetic exon-based template approach to improve identification of correctly edited alleles. The authors demonstrate its use to generate a *foxd3*CreER KI line and to engineer a single amino acid substitution in hbaa1.2. In principle, the work provides an additional toolkit for streamlining donor vector construction and improving precision in CRISPR-mediated knock-ins. The potential significance lies in easing cloning steps and providing fluorescence-based selection for editing, though the overall biological novelty is limited. In addition the use of synthetic exons for single bp substitution is not always a desirable option as changing the overall exon-intron gene organization can affect transcription. How can this applied to very long genes when the mutation is in the first exons of 10 or more? Would the authors suggest to eliminate entire kilobases of intron sequences?

SUGGESTIONS TO AUTHORS

The mini-golden system is essentially a repackaging of Golden Gate cloning for donor assembly, and its added value over existing modular strategies needs to be more convincingly demonstrated (for

instace gibson cloning). Direct quantitative comparisons to GeneWeld, cloning free approaches such as the ones proposed by Almeida et al, prime/base editing, or other recent methods are required.

The method is tested at only two loci, which is insufficient to support broad claims of generality. Additional targets should be included to provide statistical support for the efficiency and reproducibility of the approach.

Some claims in the introduction are misleading. For example, base editors are portrayed as low-efficiency tools, but multiple groups have demonstrated efficiencies approaching 100% in zebrafish and other systems. A comprehensive and up-to-date review of base editing (Liu et al., *Frontiers in Genome Editing* 2025) should be cited, and the discussion revised to more accurately reflect the state of the field. Given these advances, base editors remain the method of choice for single nucleotide substitutions when feasible.

Reply: We thank the reviewer for the thoughtful comment. We have revised our discussion to reflect recent advances in base editing technologies. We have cited the review paper as well as three more recent research papers (Zheng et al., 2023; Qin et al., 2024; Liu et al., 2025; Zhong et al., 2025). We agree with the reviewer that base editors are well suited for introducing single-nucleotide substitutions. However, in cases where base editing is not feasible, our strategy provides a practical alternative for generating amino-acid substitutions. As demonstrated in our study, *hbaa1.1* and *hbaa1.2* share over 95% genomic DNA identity, making it extremely challenging for single amino acid change using base editing. For example, identifying an appropriate sgRNA site near the target amino acid is highly difficult. In such challenging situations, particularly when no distinct sgRNA site exists nearby the target, base editing cannot be applied, and our approach offers a viable alternative. Our results provide evidence that utilizing a sgRNA site distantly located to the target site is feasible and further our strategy improves the efficiency of identifying correctly modified animals through fluorescence-based screening.

Because of the high genomic DNA similarity between the duplicated *hbaa1* genes (96% identity between *hbaa1.1* and *hbaa1.2*), base editing specifically targeting *hbaa1.2* is not feasible, and therefore we were unable to perform a direct comparison between base editing and our method. Generating and validating additional knock-in alleles for comparison would require at least two generations, each with a developmental cycle of 3 - 4 months in zebrafish. One generation is needed for injection and screening of F1 fish, and due to the low germline transmission rate, an additional generation is required to obtain a sufficient number of animals carrying the edited genome. As a result, such experiments are unlikely feasible within the revision timeline. We hope the reviewer understands these technical limitations.

We also apologize for not clearly describing the applicability of our strategy. Although our initial design resulted in replacement of a relatively large genomic region in *hbaa1.2*, our newly proposed guidelines substantially minimize the replaced segment (Fig. 6). For example, if the target amino acid is located close to the exon - intron junction, only the portion from the target codon to the nearest junction needs to be modified. This refined design allows alteration of only a small part of the exon rather than a large genomic fragment, offering greater flexibility while minimizing unintended genomic changes.

We have updated the introduction and the discussion on pages 4 and 10, respectively:
Page4

... Recent advancement of base and prime editing techniques enables the introduction of human disease-causing mutations into the zebrafish genome (Rosello et al., 2021; Petri et al., 2022; Richardson et al., 2023), significantly promoting the application of zebrafish for human disease modeling. Furthermore, iterative improvements in base editing, including optimization of the editing window, expansion of the PAM compatibilities, and introducing new DNA binding domains, have increased flexibility and broadened the range of targetable loci (Rosello et al., 2022; Zheng et al., 2023; Qin et al., 2024; Liu et al., 2025; Zhong et al., 2025). Despite these advances, the typical editing window for base editing spans several base pairs, creating the possibility of unintended edits when identical nucleotides occur within that window. Also, the current base and prime editing approaches rely on PCR-based genotyping to identify lines carrying the edited genome, which introduces multiple procedural steps and increases the risk of missing true positives. ...

Page14

... In particular, the recent advancement of base editing techniques has robustly improved efficiency (Zheng et al., 2023; Qin et al., 2024; Liu et al., 2025; Oikemus et al., 2025; Zhong et al., 2025). However, these approaches still have some limitations, including unintended conversion of bystander nucleotides and a limited set of validated conversion options. To date, C-to-T and A-to-G edits have been shown to be effective in zebrafish, whereas efficient conversion of other types has not yet been confirmed (Liu et al., 2025). ...

We have weakened our statements on page 15:

A major challenge of these approaches is how to effectively identify animals carrying the edited genome.

-> Another challenge is how to effectively identify animals carrying the edited genome.

Given the typically low efficiency of precise editing events and germline transmission rates, our method allows reliable identification of edited animals ...

-> Delete "Given the typically low efficiency of precise editing events and germline transmission rates,"

We have added new Figure 6.

Figure 6. The schematic of the proposed design for single amino acid substitution.

Reviewer 3: SUMMARY OF THE ADVANCE MADE IN THIS PAPER AND ITS POTENTIAL SIGNIFICANCE TO THE FIELD

This manuscript details a GoldenGate cloning strategy for making knockins, including Cre reporter lines and substitutions in zebrafish. The work is well documented and will be a valuable resource for the broader community. There are no major developmental insights here, but the work would fall under a techniques and resources and would be acceptable for Development.

SUGGESTIONS TO AUTHORS

The main thing I would like to see, in both examples, is a demonstration that the Cry-mCherry selection cassette can be cleanly removed with flippase and this doesn't interfere with anything. This is most critical for assessing the Cre induced expression (Figure 3) because the Cry expression could be masking ectopic Cre expression. This is even more important because the mCherry in the KI line gets dysregulated and expressed more broadly than just the lens.

It would also be important to show this in the substitution line as well; I'd especially want to see that the protein levels with and without the Cry-mCherry remain similar. The impact of this selection cassette on expression will be of interest to groups thinking about applying this method.

Reply: We appreciate the reviewer for his/her overall support and critical comments on our manuscript. In the revised manuscript, we have added new data that *flppase (flp)* mRNA injection can delete the *FRT*-flanked *a-cry:mCherry* selection cassette and addressed the reviewer's concerns.

Specific comments:

-Page 5: "Any internal Bsal sites in HAs must be mutated via overlap extension PCR or other available methods." What will that do to the efficiency of KI?

Reply: We thank the reviewer for the interesting question. Our homology arms (HAs) are approximately 400-500 base pairs (bp) in length. Previous studies have shown that even much shorter HAs (40-50 bp) can support efficient HDR under certain conditions. In our case, the 5' HA for *foxd3* contains a mutated Bsal site, yet we were still able to generate a successful KI allele. This suggests that HDR can tolerate limited sequence mismatches within the HA without substantially compromising KI efficiency. The primary goal of our manuscript is to highlight the utility of incorporating a synthetic exon and fluorescence-based screening to expand KI design strategies, and we hope the reviewer will understand that our focus is not on comparing efficiencies across

different HA configurations. We have revised the discussion.

New text on Page 13:

... Our strategy requires to mutate Bsal sites within HAs for the Golden Gate assembly. Although the impact of sequence mismatches on HDR-mediated KI is not fully understood, we successfully created the *foxd3^{CreER}* KI line despite the presence of a Bsal mutation in the 5' HA. Together with recent studies showing that short HA (40-50bp) can support efficient HDR-mediated KI (Wierson et al., 2020; Mi and Andersson, 2023; Oikemus et al., 2025), our results suggest that HDR can tolerate limited sequence mismatches within the longer HAs without substantially compromising KI efficiency. ...

-The schematic in Fig1A shows a sgRNA site and a cartoon of the primer for 5' and 3' amplification, but the logic of this is not made super clear here.

Reply: To address this comment, we have revised our supplementary data file, which contains the logic and detailed methods. We have revised the text and the figure legend to clarify that the detailed method can be found in the supplementary data.

New text on Page5 and Fig legend 1:

The detailed protocol for mini-golden system is described in **Supplementary Information**.

-Fig S1 C and D, labelling the ladder to easier understand the sizes would be helpful

Reply: We have added the corresponding size-marker ladder information to Figures 2, 5, S1, S2, S3, and S5.

-The description of the F1 generated fish in Fig 2B is confusing to follow because the text refers to fish ID numbers but only shows us a few fish, and the reader needs to keep remembering: 9= NHEJ, 23= HDR. Why not name the new alleles not with numbers but something more descriptive?

Reply: This is helpful and we now have re-labelled *foxd3^{CreER}* line9 and line 23 to *foxd3^{CreER-NH}* and *foxd3^{CreER-HR}* throughout the text and figures.

-In 2E and 2F, penetrance of the jaw defects is not large; how does that compare to homozygous LOF?

Reply: We apologize for the low number of animals tested in the initial manuscript. To address this comment, we have repeated the experiments and increased animal numbers. In addition, we have updated Figure 2, as new experiments produced images with reduced blue background, making it easier to distinguish the cartilage structure.

-Figure 3—the mCherry expression from the Cry promotor KI could be masking mis-expression of the Cre in the Ubi-switch. Test after first flipping out the Cry marker?

Reply: We thank the reviewer for motivating us to perform this experiment and include the results in this manuscript. We removed a selection maker by *flp* mRNA injection into the one-cell-stage embryos of *foxd3^{CreER-HR};ubb-switch* and examined which *foxd3^{CreER}* expressing cells recombines the *loxP* sites. Our genotyping (Fig S3) and imaging analysis (Fig. 3) demonstrated that *flp* mRNA injection successfully removed the selection marker. mCherry expression in the lens is undetectable, confirming that the mCherry signal originates from the *ubb:switch* reporter rather than from the selection marker.

New data and Figures: Fig. 3 and S3E, F

New text on page 9

... *flp* recombinase mRNA injection into the one-cell stage embryos successfully delete the *FRT*-franked *a-cry:mCherry* cassette, resulting in the lack of lens expression (Fig. 3 and S3E, F).

-Figure 4-5 The approach to use the Cry expression to screen fish for KI is interesting, but requires extensive modification of the exon-intron structure in addition to the substitution. I therefore wouldn't call it a precise substitution method, which to my mind is a change of specific bases only (maybe a few subs nearby but not artificial exons).

Reply: We agree that our synthetic exon approach alters substantial DNA sequence modifications, even though the encoded amino acids, aside from the targeted residue, remain unchanged. To

address this comment, we have changed the title and revised our text.

New title for the paper:

Advancing Knock-In Approaches for Robust Genome Editing in Zebrafish

New Sub-session title on Page 10:

Synthetic exon-based genome editing combined with fluorescence screening enables a single amino acid substitution in Hbaa1.2

Revised text on Page 10:

A primary challenge in genome editing for a.a. substitution is the low efficiency of selecting animals that carry the desired modification.

-In the expression analysis of the substitution, it would be useful to see with and without the lens marker (e.g., Flp out) to show that expression is not altered by the presence of this (or to document the extent to which is a factor)

Reply: Examining the expression level of Hbaa1.2 wild-type and Hbaa1.2 ItoV would require establishing a stable line in which the lens selection marker is removed, followed by generating homozygotes through heterozygous crosses. Because zebrafish have a generation time of 3-4 months, this process would require at least three generations, amounting to approximately one year of work. Therefore, this experiment cannot be completed within the revision timeline, and we hope the reviewer will understand this limitation. Importantly, our mass spectrometry analysis clearly detects the novel peptide corresponding to the ItoV substitution, and its abundance is higher than the wild-type peptide, supporting successful incorporation of the edited residue without compromising Hbaa1.2 expression.

First decision letter

MS ID#: bio.062437

MS Title: Advancing Knock-In Approaches for Robust Genome Editing in Zebrafish

Authors: Anjelica Rodriguez-Parks, Ella Grace Beezley, Steffani Manna, Isabella Silaban, Sarah Almutawa, Siyang Cao, Hossam Ahmed, Megan Guyer, Mark Richards, Sean Baker and Junsu Kang

I wanted to get this to you before the holidays, since I appreciate these are busy and stressful times.

Previously, you submitted your manuscript to one of our sister journals. It was reviewed, rejected, and then transferred to Biology Open. I read your manuscript carefully. I read the previous reviews carefully. I read your response to the reviewer comments and the revised manuscript carefully. I then sent you a letter explaining that your revised manuscript was still not able to be published in Biology Open because of specific points that I raised in the rejection letter. You then submitted a revised manuscript back to Biology Open, but you did not address any of the specific points that I raised. I am therefore rejecting your manuscript.

I would welcome a resubmission of your manuscript, if it addresses the points below. A revised manuscript should include a point-by-point response to the comments below, plus a version of the manuscript where any changes made in response to these comments are highlighted.

I appreciate that you responded to and revised your manuscript in light of the reviews received from the previous journal. However, in our editorial opinion, the manuscript still does not meet our rubric for publication in Biology Open. Hence, we request that you address the comments below. If after considering the feedback, you instead decide to submit elsewhere, please let me know, so that we can close our file. If you do resubmit, we offer authors a 7 working day turnaround time (decision with reviews) as part of our Fast & Fair initiative (we pay peer reviewers so that we can

provide authors with rigorous reviews, rapidly). Note that any submissions received will not be considered until Jan 5, when the holidays are over (basically, the Fast & Fair clock starts up again on Mon Jan 5). If you have any questions, don't hesitate to email me (I'll be checking email over the holidays).

Points to be addressed:

We need more clarity on the rationale. Is the rationale that your approach using the mini golden gate system offers improved efficiency over existing knockin (KI) systems in zebrafish? If so, then the introduction should clearly state the efficiency of existing systems (and discuss actual numbers of integration and germline efficiency from previous publications, not vague statements like 'Despite advancements in KI methodologies, further refinement is needed to improve precision and efficiency, enhancing the impact of genome editing on biological research'). The introduction should also describe how your results are an improvement, so that we may assess whether your results are new/an improvement in the field. Alternatively, is the rationale that base editing can be imprecise, and/or that it is difficult to screen for successful base editing mutations? If so, the intro should clearly state, and in detail, these concerns and then describe how the authors have solved this issue.

Specific examples of lack of clarity/rationale in the manuscript:

Fig 2 - *foxd3*^{CreER-HR} represents a complete *foxd3* loss-of-function allele, while *foxd3*^{CreER-NH} retains the endogenous *foxd3* activity. What is the significance of HR vs NH approaches for KI? Are the authors claiming that homology directed repair should be used to knockin Cre and knockout the target gene, while NHEJ should be used to knockin Cre while retaining target gene function? And, is it correct that, in both cases, Cre will be expressed in the same pattern as the target gene?

Fig 5 - the rationale for this figure is to improve the specificity and fidelity of base editing, but the proposed solution is to not use base editing at all, rather it is to knockin a synthetic exon containing the desired base edit. How is this different from current knockin approaches (eg, how is this a new or improved result or approach)?

Resubmission

Author response to reviewers' comments

We need more clarity on the rationale. Is the rationale that your approach using the mini golden gate system offers improved efficiency over existing knockin (KI) systems in zebrafish? If so, then the introduction should clearly state the efficiency of existing systems (and discuss actual numbers of integration and germline efficiency from previous publications, not vague statements like 'Despite advancements in KI methodologies, further refinement is needed to improve precision and efficiency, enhancing the impact of genome editing on biological research'). The introduction should also describe how your results are an improvement, so that we may assess whether your results are new/an improvement in the field. Alternatively, is the rationale that base editing can be imprecise, and/or that it is difficult to screen for successful base editing mutations? If so, the intro should clearly state, and in detail, these concerns and then describe how the authors have solved this issue.

Response

We appreciate the editor's request for clarity regarding the rationale of our approach. We apologize for any confusion and clarify here that our system advances efficiency in two distinct ways:

1) Improved donor subcloning efficiency

For knock-in (KI) genome editing, it is essential to subclone donor templates containing both 5' and 3' homology arms (HAs), requiring the assembly of multiple DNA fragments. Efficient and flexible subcloning of these donor constructs is therefore a critical but often time-consuming step. Our

mini-golden system provides a standardized and modular platform that facilitates efficient assembly of donor templates.

Among commonly used DNA assembly methods, Type IIS restriction endonuclease-based assembly (e.g., Golden Gate) and long-overlap-based assembly (e.g., Gibson assembly) are widely adopted and generally efficient. However, Gibson assembly requires long primers containing extended overlap sequences, and its efficiency can be reduced by repetitive elements or stable secondary structures. In addition, Gibson assembly is not well suited for insertion of very short DNA fragments (e.g., 20-30 nt). (Casini et al., *Nature Reviews Molecular Cell Biology* p568-576, 2015; Sorida et al., *Cell Rep Methods*. 2023 Aug 22;3(8):100564.). In contrast, Golden Gate assembly is well suited for short-fragment insertion, does not require long primers, and enables scarless and directional assembly.

Our mini golden system leverages these advantages while remaining compatible with Gibson assembly if desired. To further enhance subcloning efficiency and accessibility, we have generated and deposited 97 modular middle entry and 4 destination plasmids to Addgene, with plans to expand the middle-entry library to further streamline donor construction.

2) Improved screening efficiency for edited animals

The primary conceptual advance of our strategy lies in screening efficiency. Conventional base-editing approaches do not allow the introduction of auxiliary sequences such as FRT-flanked fluorescent selection markers. As a result, identification of correctly edited animals typically relies on PCR-based genotyping, which is labor-intensive, time-consuming, and prone to missing rare but correctly edited alleles.

In contrast, our fluorescence-based screening strategy enables rapid and robust enrichment of individual larvae that are likely to carry the edited genome. Edited animals can be identified directly by fluorescent reporter expression using simple microscopy, allowing downstream analyses to focus exclusively on enriched candidates. This approach also improves genotyping sensitivity by avoiding dilution of edited genomic DNA with unedited genomic DNA.

Additionally, our synthetic exon-based donor design enables precise introduction of amino acid substitutions while minimizing undesired recombination events, further increasing editing reliability.

Revisions to the Introduction

To address the editor's concerns, we have revised the Introduction to explicitly define these limitations and improvements. We now state:

Page 3

“Although KI strategies have been substantially optimized, editing efficiency remains variable and often low, typically ranging from 2 - 40% (Auer et al., 2014; Ata et al., 2016; Burg et al., 2018; Gutierrez-Triana et al., 2018; Prykhozhij et al., 2018; Wierson et al., 2020; Almeida et al., 2021; Levic et al., 2021; Mi and Andersson, 2023; Zhang et al., 2023; Prykhozhij and Berman, 2024; Oikemus et al., 2025). Moreover, KI for base editing generally relies on PCR-based screening to identify individuals carrying the edited genome (Rosello et al., 2022; Zheng et al., 2023; Qin et al., 2024; Liu et al., 2025; Zhong et al., 2025). However, this PCR-based screening is labor-intensive, time-consuming, and may fail to detect rare but correctly targeted events. Improving the efficiency and reliability of screening strategies is therefore essential for maximizing the practical utility of KI-based genome editing approaches.”

We have revised the third paragraph in the introduction to describe the limitation of the current base editing approach:

Page 4

“Despite these advances, the typical editing window for base editing spans several base pairs, creating the possibility of unintended edits when identical nucleotides occur within that window. Moreover, base editing approaches are inapplicable when identical sgRNA target sites are present at other genomic loci, due to the increased risk of off-target editing. Also, the current base and prime editing approaches rely on PCR-based genotyping to identify lines carrying the edited genome, which introduces multiple procedural steps and increases the risk of missing true positives. To improve screening efficiency, we incorporate a fluorescence selection marker, greatly advancing the identification of positive larvae through simple microscopy. Additionally, we employ a synthetic exon approach to avoid undesired recombination, further increasing efficiency. By integrating a synthetic exon strategy with a robust fluorescence-based screening method, we establish a highly effective approach for introducing amino acid (a.a.) changes in zebrafish models.

Our strategy offers valuable resources and methodology to the genome editing toolbox for zebrafish and other animal model communities.”

Specific examples of lack of clarity/rationale in the manuscript:

Fig 2 - $foxd3^{CreER-HR}$ represents a complete $foxd3$ loss-of-function allele, while $foxd3^{CreER-NH}$ retains the endogenous $foxd3$ activity. What is the significance of HR vs NH approaches for KI? Are the authors claiming that homology directed repair should be used to knockin Cre and knockout the target gene, while NHEJ should be used to knockin Cre while retaining target gene function? And, is it correct that, in both cases, Cre will be expressed in the same pattern as the target gene?

Response

We apologize for the lack of clarity regarding the design and intended outcome of the $foxd3^{CreER}$ alleles. Our primary goal was to generate a $foxd3$ loss-of-function allele that also functions as a CreER KI line. Specifically, we aimed to integrate CreER at the $foxd3$ start codon and to concurrently delete a portion of the $foxd3$ coding sequence. In combination with existing $foxd3$ mutants, this allele can serve as a genetic tool enabling both $foxd3$ loss-of-function analysis and CreER-mediated recombination under endogenous $foxd3$ regulatory control.

The HR-mediated KI was successfully achieved and met our goal, resulting in a bona fide $foxd3$ loss-of-function allele with CreER expressed in the endogenous $foxd3$ expression domain. By contrast, the NHEJ-mediated allele was an unintentional outcome in which $foxd3$ functionality was retained. Because this allele did not meet our experimental objective, we did not proceed with functional validation of Cre activity for the $foxd3^{CreER-NH}$ line. Importantly, we do not propose NHEJ-mediated KI as a general strategy to retain target gene function. Accordingly, our manuscript focuses on the HR-mediated $foxd3^{CreER-HR}$ allele, which fulfills our intended design criteria.

To address the editor’s comment, we have revised our text:

Page 8

“Collectively, our mini-golden-mediated strategy successfully integrated CreER into the $foxd3$ locus. $foxd3^{CreER-HR}$ represents a complete $foxd3$ loss-of-function allele, while $foxd3^{CreER-NH}$ retains the endogenous $foxd3$ activity. Because $foxd3^{CreER-HR}$ fulfilled the objective of generating both a $foxd3$ CreER driver and a loss-of-function allele, $foxd3^{CreER-HR}$ was selected for subsequent validation of Cre-mediated recombination in the NCC lineage.”

Fig 5 - the rationale for this figure is to improve the specificity and fidelity of base editing, but the proposed solution is to not use base editing at all, rather it is to knockin a synthetic exon containing the desired base edit. How is this different from current knockin approaches (eg, how is this a new or improved result or approach)?

Response

We recognize that the initial manuscript may have unintentionally conflated our strategy with base-editing approaches, and we have revised the text to clarify that our approach does not rely on base editing. Instead, it enables amino acid substitution via a synthetic exon-mediated knock-in strategy. The rationale for Figs. 4 and 5 is to provide an efficient alternative for loci that are refractory to base editing.

Base editing requires a uniquely positioned sgRNA within a narrow editing window and is not applicable when such sites are unavailable or shared across highly homologous loci. For example, $hbaa1.1$ and $hbaa1.2$ share over 95% sequence identity, and no sgRNA site proximal to the target codon uniquely distinguishes $hbaa1.2$, rendering base editing infeasible.

Our synthetic exon-mediated strategy overcomes this limitation by permitting use of a distal sgRNA site combined with homology-directed integration of a synthetic exon encoding the desired amino acid substitution. In addition, fluorescence-based selection enables robust enrichment of correctly modified animals, substantially improving screening efficiency over PCR-based approaches.

Together, these features establish our approach as a complementary and improved alternative to existing base-editing and KI strategies, particularly for highly homologous or otherwise challenging genomic loci.

To address the editor’s comment, we revised the text by reordering the relevant sections and adding clarifying language.

Page 10 - Introductory paragraph of the section entitled “Synthetic exon-based genome editing combined with fluorescence-based screening enables precise single-amino acid substitution in $Hbaa1.2$,” which introduces the rationale for using a synthetic exon strategy to overcome limitations of base-editing approaches at highly homologous genomic loci.

“*hbaa1.1* and *hbaa1.2* exhibit extremely high sequence identity at the a.a., coding, and genomic DNA levels (Fig. 4B). Although most sgRNA target sites are shared between the two genes, we identified an efficient sgRNA site specific to *hbaa1.2*, which is positioned 63 base pairs upstream of ATC (I64) codon. This site exhibits two nucleotide differences relative to the corresponding region of *hbaa1.1* (Fig. 4C, D). Because the sgRNA site is distal to the target IleE11 codon, the base editing approach is inapplicable for introducing the Hbaa1.2 ItoV substitution.

Another challenge for a.a. substitution ...”

First decision letter

MS ID#: bio.062472

MS Title: Advancing Knock-In Approaches for Robust Genome Editing in Zebrafish

Authors: Anjelica Rodriguez-Parks, Ella Grace Beezley, Steffani Manna, Isabella Silaban, Sarah Almutawa, Siyang Cao, Hossam Ahmed, Megan Guyer, Mark Richards, Sean Baker and Junsu Kang

Thank you for responding to my comments and revising the manuscript accordingly. I am happy to tell you that your manuscript has been accepted for publication in Biology Open, pending our standard publication integrity checks. It was accepted on 14th January 2026.